# Polymorphism of pyrene on compression to 35 GPa in a diamond anvil cell
Wenju Zhou [1] ✉, Yuqing Yin[2], Dominique Laniel[3], Andrey Aslandukov [1,4], Elena Bykova [5], Anna Pakhomova[6], Michael Hanfland[6], Tomasz Poreba [6], Mohamed Mezouar [6], Leonid Dubrovinsky [4] & Natalia Dubrovinskaia [1,2] ✉

Structural studies of pyrene have been limited to below 2 GPa. Here, we report on investigations of pyrene up to ~35 GPa using in situ single-crystal synchrotron X-ray diffraction in diamond anvil cells and ab initio calculations. They reveal the phase transitions from pyrene-I to pyrene-II (0.7 GPa), and to the previously unreported pyrene-IV (2.7 GPa), and pyrene-V (7.3 GPa). The structure and bonding analysis shows that gradual compression results in continuous compaction of molecular packing, eventually leading to curvature of molecules, which has never been observed before. Large organic molecules exhibit unexpectedly high conformational flexibility preserving pyrene-V up to 35 GPa. Ab initio calculations suggest that the phases we found are thermodynamically metastable compared to pyrene-III previously reported at 0.3 and 0.5 GPa. Our study contributes to the fundamental understanding of the polymorphism of polycyclic aromatic hydrocarbons and calls for further theoretical exploration of their structure–property relationships.

Polycyclic aromatic hydrocarbons (PAHs) have long attracted interest as potential materials for various optical, optoelectronic, and electronic applications[1–3]. In addition to application-oriented research, much work has been done for understanding the fundamental processes associated with their structure–property relationships. For example, the electronic and excitonic processes in aromatic crystals have been strongly linked to both the number of aromatic rings in the molecular structure and the arrangement of molecules in the crystal[1,2].

Pressure has been proven to be a very powerful thermodynamic parameter that induces structural transformations affecting materials' properties, so exploring the behavior of PAHs under pressure may provide insights into the structural transitions and intermolecular interactions for this important class of organic materials. However, so far, the information about the structural behavior of any organic crystals at pressures exceeding a few gigapascals is very limited. It is mainly due to studies at higher pressures having been hindered by both the technical complexity of the experiments on fragile organic crystals and because of a common belief that the crystals are quickly destroyed under compression. The advancement of single-crystal X-ray diffraction (SC-XRD) techniques in diamond anvil cells (DACs) with the use of soft pressure-transmitting media[4], such as inert gases, has created new possibilities for investigating crystal structures, phase

transitions, equilibrium and non-equilibrium transformation paths, molecular arrangements, and chemical bonding of organic crystals under previously unexplored high-pressure conditions. For example, recently high-pressure polymorphism in L-threonine was studied between ambient pressure and 22 GPa[5]. Nevertheless, large molecules of PAHs have long been supposed to have low conformational flexibility under pressure and so far have been studied using SC-XRD only up to 2.1 GPa[6].

Pyrene ($C_{16}H_{10}$) is a representative of PAHs. At ambient conditions, it is a solid with a monoclinic structure ($P2_1/c$ space group)[7] called pyrene-I. Flat pyrene molecules made of four fused benzene rings form pairs ("sandwiches") packed in a herringbone motif (Fig. 1a). Because of such molecular packing and the ring's conjugated $\pi$-system, pyrene crystals are of interest to study under pressure to examine pressure-induced structural transformations, changes in packing of molecular units, and chemical bond evolution in PAHs. Previous structural studies of pyrene, using diffraction methods, enabled to establishment of its two polymorphs. The first polymorph, pyrene-II (Fig. 1b), was identified upon a transition from pyrene-I at low temperature[8–10]. Its structural motif is similar to that of pyrene-I. The other polymorph, pyrene III (Fig. 1c) was identified on single crystals of pyrene recrystallized from a dichloromethane solution at 0.3 GPa and 0.5 GPa[6]. It was found to have a different molecular packing model and

[1]Material Physics and Technology at Extreme Conditions, Laboratory of Crystallography, University of Bayreuth, Bayreuth, Germany. [2]Department of Physics, Chemistry and Biology (IFM), Linköping University, Linköping, Sweden. [3]Centre for Science at Extreme Conditions and School of Physics and Astronomy, University of Edinburgh, Edinburgh, UK. [4]Bayerisches Geoinstitut, University of Bayreuth, Bayreuth, Germany. [5]Institut für Geowissenschaften, Goethe-Universität Frankfurt, Frankfurt am Main, Germany. [6]European Synchrotron Radiation Facility, Grenoble, France. ✉e-mail: Wenju.Zhou@uni-bayreuth.de; Natalia.Dubrovinskaia@uni-bayreuth.de

**Fig. 1 | Crystal structures of pyrene polymorphs viewed along the *a*-axis. a** Pyrene-I, ambient conditions; **b** pyrene-II, 0.7 GPa; **c** pyrene-III, 0.5 GPa[6]; **d** pyrene-IV, 2.7 GPa; and **e** pyrene-V, 7.3 GPa. Pairs of molecules (sandwiches) determining the molecular packing sandwich-herringbone motif in pyrene-I, pyrene-II, and pyrene-IV are shown at the bottom. Pyrene-IV has two types of sandwiches (sandwich 1—to the left, sandwich 2—to the right). Pyrene-III and Pyrene-V do not feature sandwiches. C atoms are black, and H atoms are white.

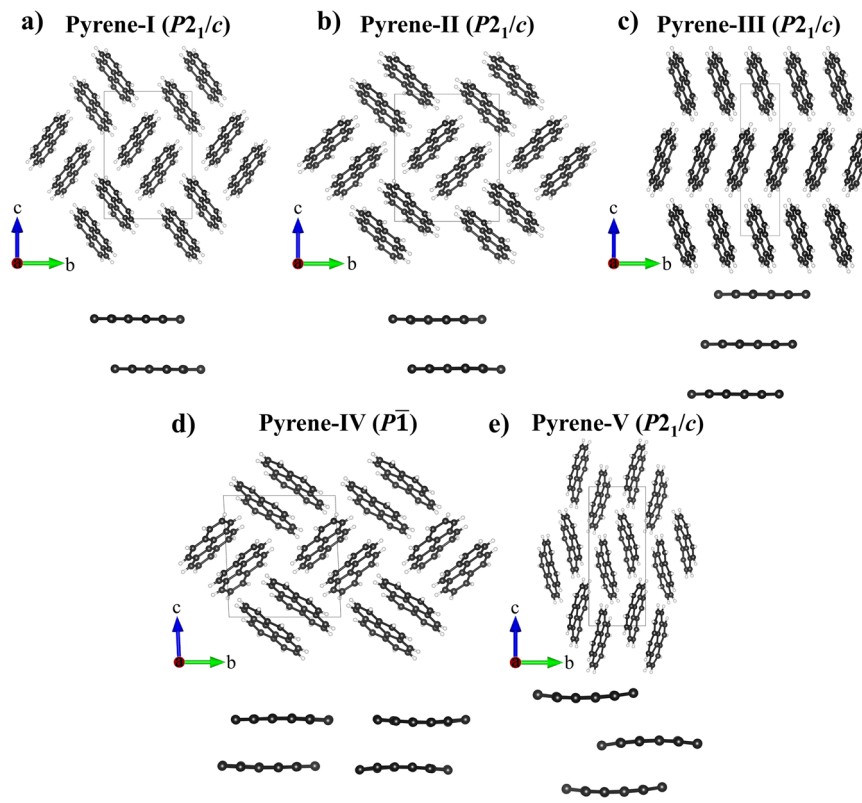

different intermolecular interactions. The structures will be discussed in detail below in relation to our findings.

Although spectroscopic data do not provide explicit information about the structure of solid matter, it is worth noticing that vibrational spectroscopy investigation of pyrene up to about 1 GPa, pointing towards the existence of phase transformations in pyrene under pressure, was made as early as in 1976[11], when a transition was detected on an abrupt change of the Raman spectrum at ca. 0.4 GPa. Later Raman spectroscopy study[12] detected a transformation at 0.3 GPa on a crystal grown from a dichloromethane solution, and on a crystal pressurized in argon up to 0.6 GPa, interpreted in both cases as observation of pyrene-III, similar to that described by Fabbiani et al.[6].

In this work, we have investigated the behavior of pyrene in the pressure range from ambient to 35.5 GPa using synchrotron SC-XRD in DACs. Supplementary Table 1 provides a summary of all experiments. We have observed three high-pressure polymorphs: pyrene-II, whose structure was known from low-temperature experiments at ambient pressure, but we solved and refined it at high pressure for the first time, and two new phases, pyrene-IV and pyrene-V. Here we report the results of the analysis of their structures and bonding evolution under pressure. Contrary to the previous belief that large organic molecules have low conformational flexibility under pressure, we have demonstrated that gradual compression results in continuous compaction of molecular packing, eventually leading to curvature of molecules, which has never been observed before. Our results reveal that pyrene-V can be preserved in He pressure medium up to ~35 GPa due to fully unexpected structure compaction accompanied by considerable deformation of molecules and their strong alignment along one crystallographic axis.

## Results and discussion
### Crystal structures of pyrene polymorphs

Upon compression of pyrene-I ($P2_1/c$) to 0.7 GPa in a He pressure medium, we observed a phase transition to pyrene-II ($P2_1/c$) (Fig. 1b), which was still preserved at 1.4 GPa. At the next pressure step (2.7 GPa), a previously unknown triclinic polymorph of pyrene, pyrene-IV ($P\bar{1}$) (Fig. 1d), was identified. The next phase transition occurred at 7.3 GPa to pyrene-V

(Fig. 1e) with a monoclinic structure ($P2_1/c$). Further, we describe in detail the structures of all polymorphs observed in this work. Full crystallographic and experimental data are provided in Supplementary Tables 2 through 5 and Supplementary Data 1 through 12.

The crystal structure of pyrene-I (Fig. 1a) was first reported in the SC-XRD study of Robertson and White in ref. 13. It was later refined by means of neutron diffraction (CSD reference code PYRENE02)[14]. The crystallographic data of pyrene-I obtained in this work based on synchrotron SC-XRD in comparison with neutron diffraction data of Hazell et al.[14] are provided in Supplementary Table 2. The structure is monoclinic (space group #14, $P2_1/c$) with the following unit cell parameters at ambient conditions: $a = 8.478(8)$ Å, $b = 9.2562(12)$ Å, $c = 13.655(7)$ Å, $\beta = 100.31(8)°$ and $V = 1055.3(11)$ Å$^3$.

A transition from pyrene-I to pyrene-II below 110 K was first reported by Jones et al.[8], and the structure of pyrene-II was suggested based on a combination of micro-electron diffraction and atom—atom, pairwise potential calculations. Knigt et al.[9] confirmed and refined the pyrene-II structure from high-resolution neutron powder diffraction data collected from a fully deuterated sample at 4.2 K. First single-crystal XRD analysis of pyrene-II at 93 K and ambient pressure were reported by Frampton et al.[10]. Our work reports the first structural analysis of pyrene-II under pressure at room temperature using single-crystal XRD and provides crystallographic data for pyrene-II, which are in a very good agreement with those obtained at low temperature and ambient pressure[8–10].

We observed pyrene-II and solved and refined its structure at two pressure points (0.7 GPa and 1.4 GPa) upon pressurizing crystals of pyrene-I in a helium pressure medium; see Supplementary Table 3 for our crystallographic data in comparison with the low-temperature data of ref. 10. It has the same space group as pyrene-I with the following unit cell parameters at 0.7 GPa: $a = 8.1431(12)$ Å, $b = 9.8639(7)$ Å, $c = 12.1136(4)$ Å, $\beta = 96.484(7)°$ and $V = 966.77(16)$ Å$^3$, and a similar sandwich-herringbone molecular packing motif if viewed along the *a* direction. The $\beta$ angle in pyrene-II is about 4° larger than in pyrene-I. Upon compression, it slightly decreases.

The structures of the two polymorphs, pyrene-I and pyrene-II (Fig. 1a, b), are very similar. As underlined in previous studies[8,10], "a small

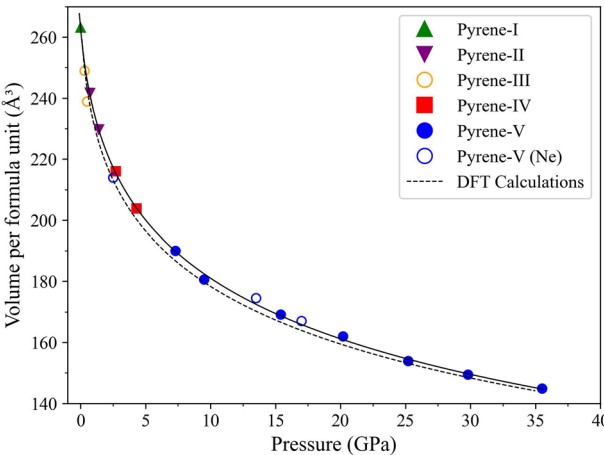

**Fig. 2 | Compressional behavior of the polymorphs of pyrene up to 35.5 GPa.** The unit cell volume per formula unit as a function of pressure is presented for pyrene-II (purple solid inversed triangles), pyrene-IV (red solid squares), and pyrene-V (blue solid circles) as found in this work in experiments with He pressure medium. The green solid triangle corresponds to pyrene-I at ambient conditions. The solid black line shows the fit of all pressure–volume experimental points using the third-order Birch–Murnaghan EOS with the parameters $V_0 = 263.8(4)$ Å$^3$, $K_0 = 5.2(2)$ GPa, and $K' = 10.6(4)$ (the EoSFit7 software was utilized). Open blue circles correspond to the pressure–volume data for pyrene-V in aNe pressure medium. Open orange circles correspond to the data for pyrene-III from ref. 6. The dashed line presents the result of the fit of the DFT calculated pressure–volume points for each polymorph in the pressure interval in which these phases were observed experimentally. The fit parameters are as follows: $V_0 = 261.0(8)$ Å$^3$, $K_0 = 3.4(2)$ GPa, and $K' = 15.2(7)$.

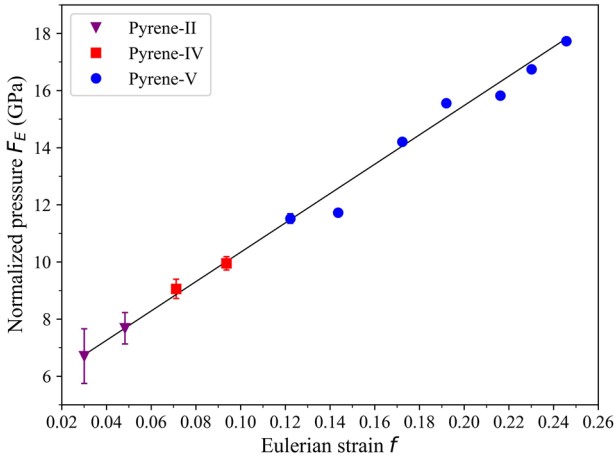

**Fig. 3 | The Eulerian strain–normalized pressure (f–F) plot of the experimental data.** In our experimental data, pyrene-II is depicted by purple inverted solid triangles, pyrene-IV is denoted by red squares, and pyrene-V is represented by blue solid circles. The solid line represents the linear fit.

## Compressional behavior of the polymorphs of pyrene

The compressional behavior of the polymorphs of pyrene up to 35.5 GPa is presented in Fig. 2. Further pressurization led to the loss of the XRD signal. The values of the unit cell volume per formula unit for pyrene-I at ambient conditions and for pyrene-II, pyrene-IV, and pyrene-V as a function of pressure (in He pressure medium) were obtained from our experiments (Supplementary Table 6). In Fig. 2 they are shown by solid symbols of different colors. These pressure–volume data were fitted using the third-order Birch–Murnaghan equation of state (EOS) with the fixed zero-pressure volume $V_0 = 263.8$ Å$^3$, which is the volume of pyrene-I at ambient conditions. The bulk modulus, $K_0$, and its first derivative, $K'$, were determined to be 5.2(2) GPa and 10.6(4) (Figs. 2 and 3) using EoSFIT7 software[15]. The pressure–volume points for pyrene-V in Ne pressure medium, like also those for pyrene-III from ref. 6, were not included in the fit, although they are shown in the figure.

The density functional theory (DFT) calculated pressure–volume points for each polymorph (Supplementary Table 7) in the pressure interval, in which these phases were observed experimentally, have also been fitted using the third-order Birch–Murnaghan EOS. The EOS parameters appeared to be as follows: $V_0 = 261.0(8)$ Å$^3$, $K_0 = 3.4(2)$ GPa, and $K' = 15.2(7)$. They agree well with the EOS parameters obtained from the experimental data (Fig. 2).

The dependence of the lattice parameters of pyrene polymorphs on pressure is shown in Fig. 4 (see Supplementary Table 8 for numerical values). In each polymorph, all parameters gradually decrease in response to pressurization. Phase transitions are manifested by abrupt changes in particular parameters. Whereas the $a$ parameter always shortens (Fig. 4a), the value of b (Fig. 4b) increases upon the transition from pyrene-I to pyrene-II and from pyrene-II to pyrene-IV, but sharply decreases upon the transition from pyrene-IV to pyrene-V. The value of the c parameter (Fig. 4c) decreases upon the transition from pyrene-I to pyrene-II and from pyrene-II to pyrene-IV, but sharply increases (from 10.9 Å to 16.1 Å) upon the transition from pyrene-IV to pyrene-V. The variation in the $\beta$ angle (Fig. 4d) shows a decrease from 100.3° in pyrene-I to 96.5° in pyrene-II, further reducing to 95.0° in pyrene-IV, and then increasing to the value of 100.7° in pyrene-V, similar to that in pyrene-I. Although the pressure–volume points of pyrene-III[6] fit the pressure–volume curve of other polymorphs (Fig. 2), the unit cell parameters of pyrene-III are quite different from those of pyrene-I and pyrene-II.

To summarize, we could describe the P–V behavior for all pyrene polymorphs by a common continuous EOS, but in fact, the volume may change continuously with pressure, while other structural parameters abruptly change manifesting solid state transitions (Figs. 2 and 4).

rotation of molecules around the $c$-axis [it corresponds to the $a$-axis in the standard setting $P2_1/c$ used in our paper for space group #14] of the pyrene-I unit cell generates a new structure that is very close in terms of cell dimensions and packing motif to pyrene-II" (cited from ref. 10). Namely this rotation is responsible for considerable change in the molecules interplanar angle (see the analysis below).

Further compression of pyrene-II led to the formation of a previously unknown triclinic polymorph of pyrene, pyrene-IV (space group #2, $P$-1), which we observed at 2.7 GPa and 4.3 GPa. The unit cell parameters at 2.7 GPa are as follows: $a = 7.593(3)$ Å, $b = 10.223(3)$ Å, $c = 11.192(2)$ Å, $\alpha = 92.536(19)°$, $\beta = 100.31(8)°$, $\gamma = 91.21(3)°$, and $V = 864.2(5)$ Å$^3$ (Fig. 1d). Supplementary Table 4 provides the crystallographic data for pyrene-IV at 2.7 GPa and 4.3 GPa. In pyrene-IV, there are two crystallographically different molecules forming two kinds of sandwiches, one consisting of almost flat molecules and another of curved ones.

Strictly speaking, pyrene-IV does not possess the herringbone motif anymore, due to reducing the symmetry down to $P$-1 and loss of the $pgg$ symmetry in the projection along the $a$-axis. However, due to the angles $\beta$ and $\gamma$ being so close to 90°, it is practically invisible (Fig. 1d). So we can say that compression up to 4.3 GPa doesn't change the molecular packing motif viewed along the $a$-axis, it still remains sandwich-herringbone-like.

A new polymorph, pyrene-V, was first observed at 7.3 GPa (Fig. 1e). It has the same space group as pyrene-I and pyrene-II (space group #14, $P2_1/c$) with the following unit cell parameters at 7.3 GPa: $a = 7.450(5)$ Å, $b = 6.4503(12)$ Å, $c = 16.096(2)$ Å, $\beta = 100.65(3)°$, and $V = 760.1(5)$ Å$^3$. Supplementary Table 5 provides detailed crystallographic data of pyrene-V for six pressure points in the range of 7.3 GPa to 35.5 GPa.

As seen (Fig. 1e), the compression leads to the collapse of the sandwich structure in pyrene-V. The crystallographically equivalent molecules are aligned at a very low angle, they are substantially curved and shifted with respect to each other, forming a simple herringbone packing motif. A detailed geometrical analysis of the structures and shapes of molecules in different polymorphs is given in a separate section below.

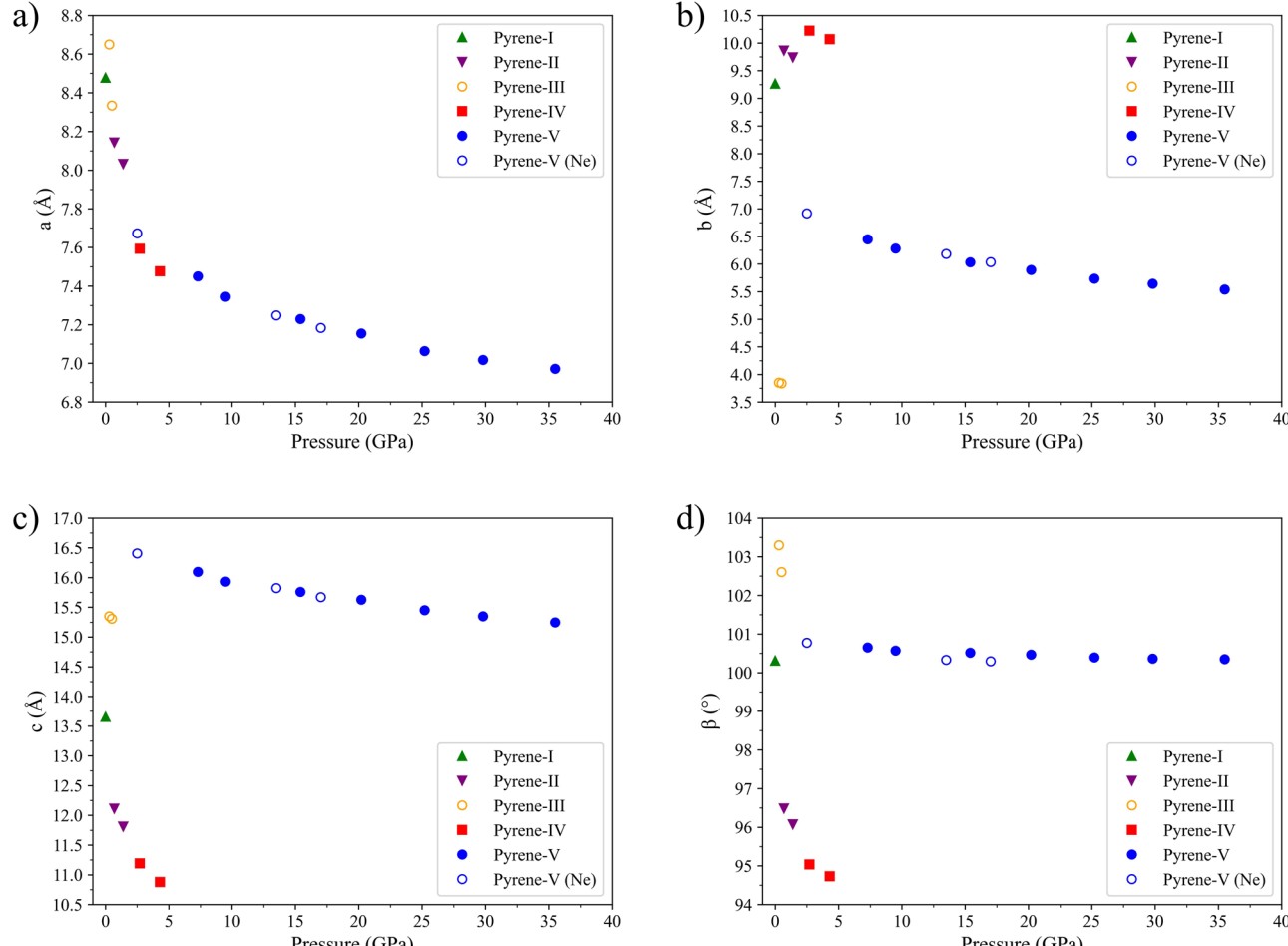

**Fig. 4 | The dependence of the lattice parameters of pyrene polymorphs on pressure. a** $a$; **b** $b$; **c** $c$; and **d** $\beta$. Solid symbols correspond to the data points for the samples measured in He pressure transmitting medium: pyrene-I (green triangles), pyrene-II (purple inversed triangles), pyrene-IV (red squares), and pyrene-V (blue circles). Open blue circles are for pyrene-V in Ne pressure medium. Open orange circles correspond to the data for pyrene-III from ref. 6. The pyrene-I-to-pyrene-II transition manifests in an increase of the $b$ parameter and decrease of the $\beta$ angle; for the pyrene-II-to-pyrene-IV transition—in the decrease of the symmetry and increase of the $b$ parameter; and for the pyrene-IV-to-pyrene-V – in the increase of the symmetry and the abrupt decrease of the $b$ parameter and increase of the $\beta$ angle.

## Theoretical calculations

The relaxed structural parameters of pyrene-I at ambient, pyrene-II at 1 GPa, pyrene-III at 1 GPa, pyrene-IV at 3 GPa, and pyrene-V at 9 GPa are provided in Supplementary Tables 9–13. The calculated unit cell volumes are slightly smaller than the experimental ones, likely due to the impact of temperature (0 K) on the results of calculations.

The enthalpy differences ($\Delta H$) for the four polymorphs (pyrene-II, pyrene-III, pyrene-IV, and pyrene-V) relative to pyrene-I were calculated as a function of pressure up to 5 GPa at 0 K (Supplementary Table 14 and Fig. 5), as described in the Methods section. The calculations suggest that up to 2.07 GPa pyrene-II is relatively more stable than pyrene-IV and pyrene-V. We observed its formation at 0.7 GPa and 1.4 GPa in our room temperature experiment which agrees with the calculations. Above 2.07 GPa pyrene-V is predicted to be more stable than other polymorphs except pyrene-III. This does not contradict to our observations, as we detected pyrene-V formation at 7.3 GPa at the pressure step from 4.3 GPa. It is known that the formation of metastable phases is very sensitive to many parameters like stress, for example, which cannot be fully controlled in a DAC experiment, as we have shown previously in our work on high-pressure phases of silica[16].

As seen in Fig. 5, above 0.03 GPa and up to 5 GPa, pyrene-III appears to be the thermodynamically stable phase if compared to all other polymorphs. In our room temperature experiments, we did not observe pyrene-III described in the study of Fabbiani et al.[6], where it was synthesized through the recrystallization from a 0.5 M solution of pyrene in dichloromethane after several temperature-annealing cycles (slow cooling and heating between 303 K and 293 K) at 0.3 GPa. This work and our computational result motivated us to conduct a high-pressure high-temperature experiment. This experiment was designed as described below.

A sample of pyrene-I was loaded into a DAC along with KCl as a pressure-transmitting medium (DAC #4). The whole DAC (first pressurized to 4 GPa) was heated for two hours in an oven at 473 K. After heating pressure was raised to 6.5 GPa and at this pressure, the sample was investigated at the ID15B beamline at the ESRF ($\lambda = 0.4100$ Å). Dioptas program[17] and Jana2006 program[18] were used for the data processing. The EOS of KCl was adopted from[19]. The diffraction pattern of the sample featured SC-XRD reflections of two pyrene polymorphs (pyrene-III and pyrene IV) and continuous diffraction lines of the KCl pressure medium (Fig. 6). This observation suggests that pyrene-III is likely a thermodynamically stable phase at pressures above 0.3 GPa, whose synthesis requires heating to overcome the energy barrier.

Upon compression at room temperature, the visual appearance of the pyrene crystals changes: from colorless and transparent they become orange and eventually black (Fig. 7). Calculations of the electronic density of states (eDOS) for pyrene-I at 1 bar and 0 K and pyrene-V at 50 GPa and 0 K (Fig. 8) have shown that the band gap decreases from 3.3 eV to 0.9 eV. This explains the observed color change of the crystals.

## Geometrical analysis of the structures of the pyrene polymorphs

In order to accurately calculate the distances between pyrene molecules in pairs (sandwiches), pyrene molecules were approximated by mean molecular planes by performing planar fitting for 16 carbon atoms of pyrene molecules in each polymorph (Fig. 9) using the NumPy and SciPy libraries in Python. Blue lines in Fig. 9 highlight the mean molecular planes; the intermolecular distances (d) and interplanar angles ($\delta$), which were determined using the same software, are designated. In Fig. 9 the structures of pyrene-I, II, and IV are shown along the [4 0 1] direction (Fig. 9a–c), and that of pyrene-V in the [5 0 2] direction (Fig. 9d). The selection of these two specific orientations enables the best view of the topology of the molecular structures of the different polymorphs.

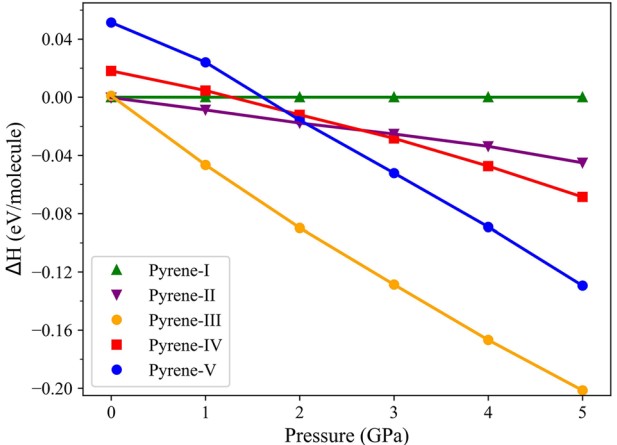

**Fig. 5 | The enthalpy difference ($\Delta H$) calculated for the four polymorphs (pyrene-II, pyrene-III, pyrene-IV, and pyrene-V) relative to pyrene-I as a function of pressure.** All calculations were performed at 0 K. The data for different polymorphs are shown as follows: pyrene-I (green triangles); pyrene-II (purple inversed triangles), pyrene-III (orange circles), pyrene-IV (red squares), and pyrene-V (blue circles). Up to 2.07 GPa, pyrene-II is relatively more stable than pyrene-IV and pyrene-V, whereas above this pressure, pyrene-V is relatively more stable. Above 0.03 GPa and up to 5 GPa, pyrene-III appears to be the thermodynamically stable phase if compared to all other polymorphs.

Intermolecular distances and interplanar angles in pyrene polymorphs from experiments in the helium pressure medium are listed in Supplementary Table 15 (for graphical representation of their pressure dependences see Fig. 10). As seen, the phase transitions from pyrene-I to pyrene-II and then to pyrene-IV result in the formation of more compact structures (with similar sandwich-herringbone molecular packing) due to a general shortening of intermolecular distances and decrease of interplanar angles (from 83.6° to 75.4°, and 66.7° in pyrene-I, pyrene-II, and pyrene-IV, respectively) (Supplementary Table 15 and Fig. 10), as well as due to a mutual shift of the molecules in sandwiches (Fig. 9). In pyrene-IV such compaction eventually leads to reduction of the symmetry to $P$-1, and the appearance of two crystallographically distinct molecules forming pairs with different intermolecular distances ($d_1 < d_2$, Figs. 9c and 10a). Such a "tension" in the structure of pyrene-IV is "released" in pyrene-V, which, like pyrene-I and pyrene-II, features crystallographically equivalent molecules. The latter, however, do not form pairs anymore, but shift with respect to each other and tightly align along the $c$ direction (Fig. 1e) upon a drastic decrease of the $\delta$ angle down to 44.9° (Figs. 9d and 10b) that leads to the collapse of the sandwich structure. The symmetry of pyrene-V, compared to that of pyrene-IV, increases to $P2_1/c$. Interestingly, although twisted molecules in organic solids at ambient conditions are known (see for example, ref. 20), to the best of our knowledge, the phenomenon of the increase of the curvature upon gradual compression is reported here for the first time.

Due to all molecules in pyrene-V being curved, there are two types of contacts between the molecules: each molecule has in its proximity both a convex and a concave neighbor at the distances of $d_1$ and $d_2$, respectively, so that $d_1 > d_2$. Such a geometrical arrangement results in a sharp raising of the $c$ parameter (from 10.9 Å in pyrene-IV to 16.0 Å in pyrene-V) (Supplementary Table 8 and Fig. 4c) and a decrease of the $b$ parameter (from 10.1 Å to 6.5 Å) (Fig. 4b), accompanied by a considerable expansion of the $\beta$ angle to 100.5° (Supplementary Table 8 and Figs. 4d and 9d) of the unit cell of pyrene-V.

To visualize changes in the curvature of the pyrene molecules under increasing pressure, we performed curved surface fitting using the NumPy and SciPy libraries in Python. We set the normal vector to the mean molecular plane described above. The height of the curved molecular surface, represented by the projection of each point on the normal axis, directly reflects the degree of curvature and the shape of the surface.

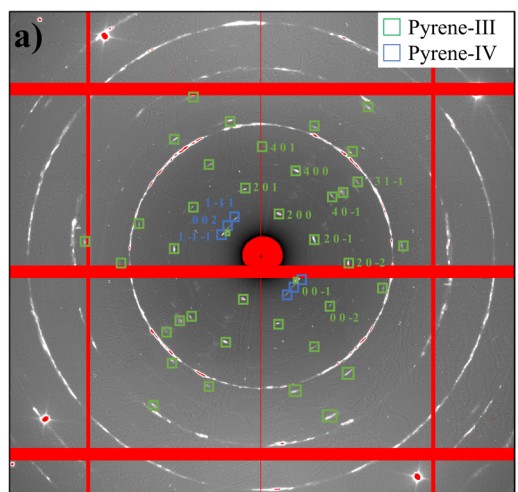

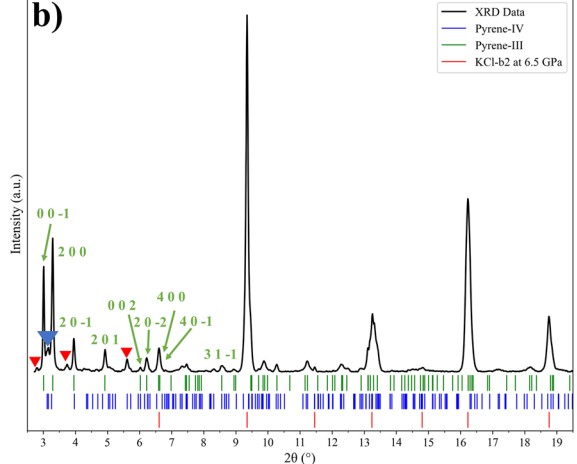

**Fig. 6 | The XRD pattern obtained from a sample of pyrene in a KCl pressure medium at 6.5 GPa and room temperature after it was heated at 473 K for 20 min. a** A 2D XRD pattern resulted from merging individual frames obtained upon an omega scan (±34° with a step of 0.5°) using the Dioptas program[17]. The diffraction spots from pyrene-III are highlighted by green boxes and those from pyrene-IV—by blue boxes. Continuous white circles are from KCl. **b** Integrated X-ray diffraction pattern ($\lambda = 0.410$ Å). Green ticks correspond to the positions of diffraction lines of pyrene-III at 6.5 GPa ($a = 14.596(5)$ Å, $b = 3.463(3)$ Å, $c = 7.981(3)$ Å, $\beta = 102.39(3)°$, and V = 394.0(7) Å$^3$, lattice parameters refinement was performed using the Jana2006 program[18]); blue ticks—those of pyrene-IV ($a = 7.467(6)$ Å, $b = 10.057(6)$ Å, $c = 10.865(7)$ Å, $\alpha = 92.43°$, $\beta = 94.73°$, $\gamma = 90.86°$ and V = 812.2(15) Å$^3$, all angles are fixed at the values of pyrene-IV at 4.3 GPa in this work); red ticks—those of B2-KCl[19] at 6.5 GPa. The peak marked with a blue solid inversed triangle results from overlapping of 002, 1–1–1, and 1–11 reflections of pyrene-IV. The peaks marked with red triangles are from non-identified spots.

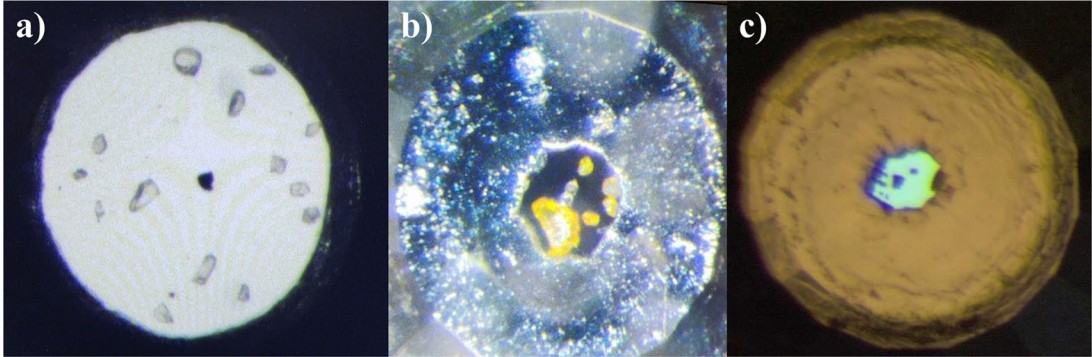

**Fig. 7 | Images of pyrene crystals in DACs taken under an optical microscope at different pressures. a** Numerous colorless transparent crystals of pyrene-I at ambient pressure (DAC#1), the black triangular feature is due to a piece of tungsten used for the cell alignment in the X-ray beam); **b** light orange crystals of pyrene-V at 2.5 GPa (DAC#2); **c** non-transparent dark crystals of pyrene-V at ~35 GPa (DAC#3). The average size of the crystals is about 10 μm.

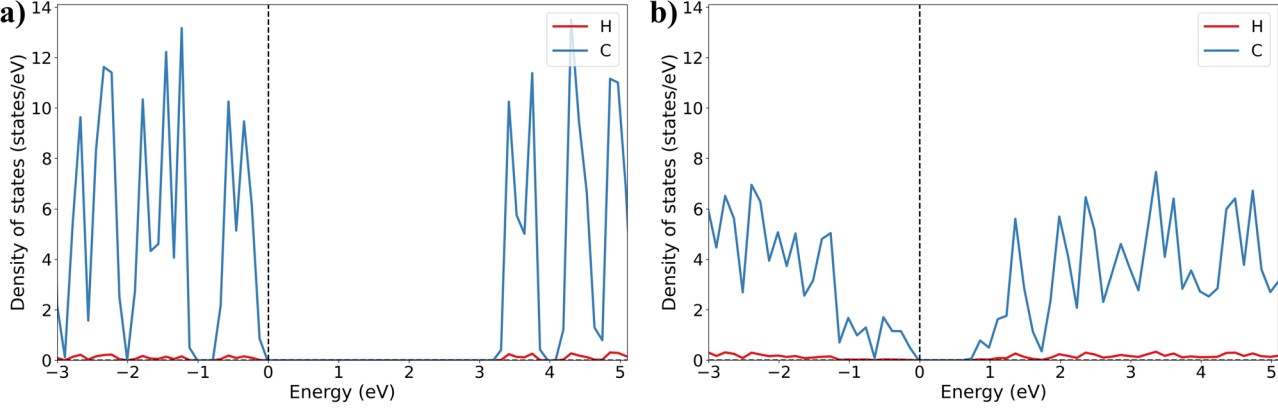

**Fig. 8 | Electronic densities of states of pyrene polymorphs. a** Pyrene-I at 1 bar; **b** pyrene-V at 50 GPa. The partial density of states projected onto the C atoms is shown by blue curves, H atoms are shown by red curves, and the Fermi energies—by the vertical dashed lines. The band gap of pyrene-I is 3.3 eV at ambient conditions and the band gap of pyrene-V is 0.9 eV at 50 GPa.

**Fig. 9 | Visualization of intermolecular distances and interplanar angles in the structures of pyrene polymorphs. a** Pyrene-I, **b** pyrene-II, and **c** pyrene-IV, as viewed along the [4 0 1] direction, and **d** pyrene-V as viewed along the [5 0 2] direction. The blue lines represent the mean molecular planes determined by the positions of carbon atoms in the molecules; $d_1$ and $d_2$ are the interplanar distances; $\delta$ is the interplanar angle. In pyrene-IV $d_1 < d_2$; in pyrene-V $d_1 > d_2$. C atoms are black. Hydrogen atoms are not shown.

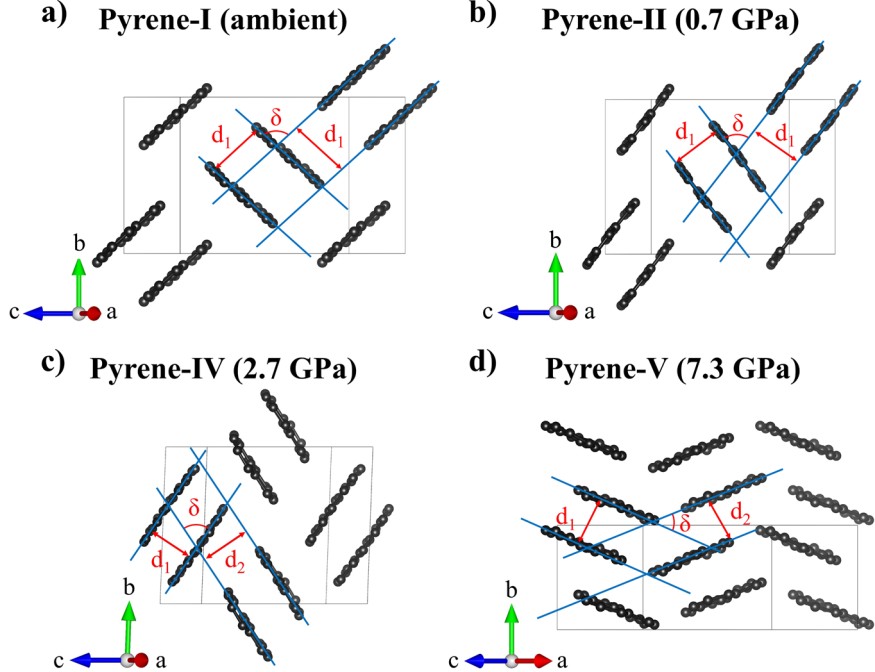

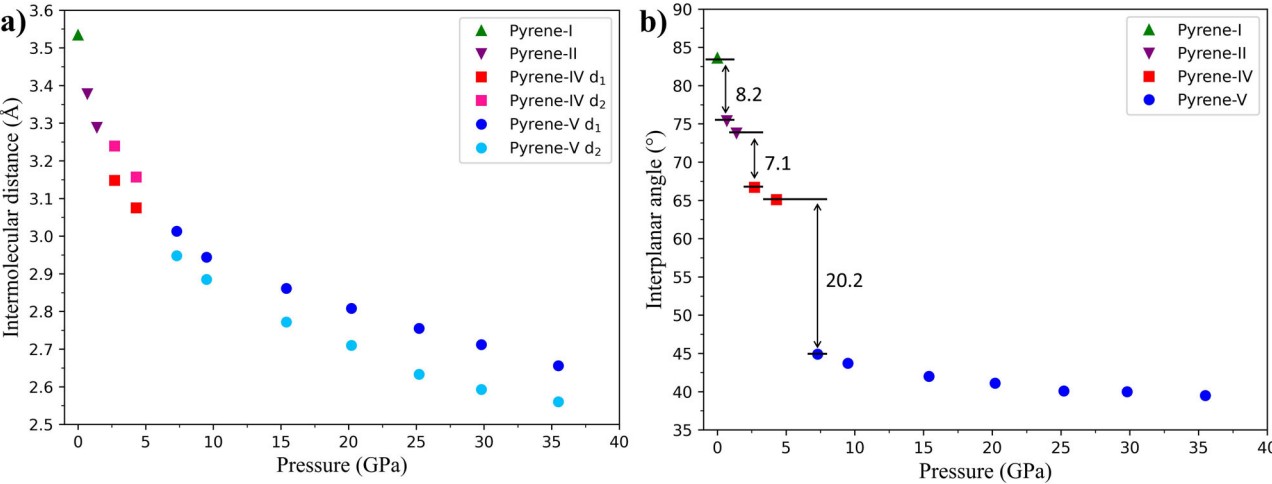

**Fig. 10 | Variation of intermolecular distances and interplanar angles in pyrene polymorphs with pressure. a** Intermolecular distances see (Fig. 5 for $d_1$ and $d_2$); **b** interplanar angles (see Fig. 5 for δ). Only the data obtained from pyrene in He pressure medium are presented.

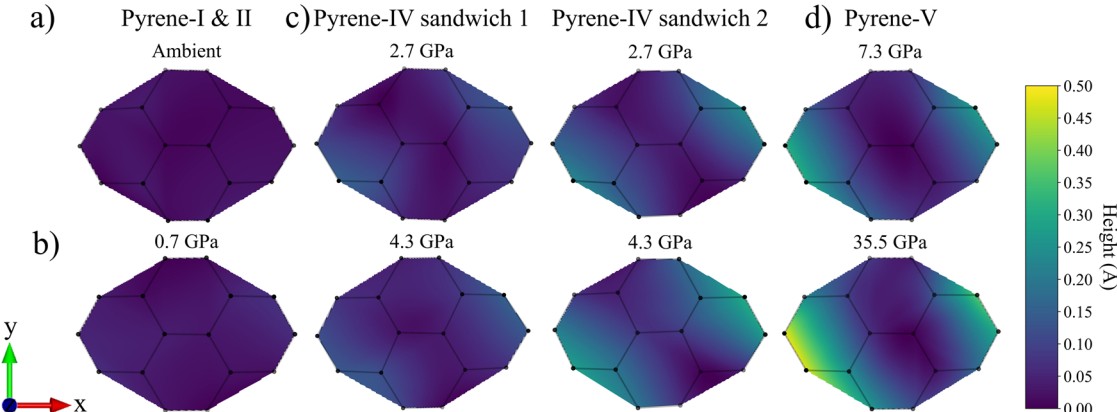

**Fig. 11 | Curved surface plots for pyrene molecules in different polymorphs at selected pressures. a** Pyrene-I at ambient pressure; **b** pyrene-II at 0.7 GPa; **c** pyrene-IV at 2.7 GPa and 4.3 GPa with sandwich 1 and sandwich 2; and **d** pyrene-V at 7.3 GPa and 35.5 GPa. The curved surface plots visualize the distribution of surface height variations, represented as the difference between the height at any given point on the surface and the minimum height of the surface. The range of height differences spans from 0 Å to 0.5 Å, with a corresponding color gradient ranging from purple to yellow (the color scale is given to the right of the figures). The plots were produced using matplotlib and NumPy libraries in Python. C atoms are black spots.

The obtained curved surface plots (Fig. 11) visualize the distribution of surface height variations, represented as the difference between the height at any given point on the surface and the 0 Å (minimum) height of the surface. The range of height differences spans from 0 Å to 0.5 Å, with a corresponding color gradient ranging from purple to yellow. Figure 12a shows the pressure dependence of the maximal surface height difference for all studied polymorphs.

As seen in Fig. 11a, b, molecules of pyrene-I and pyrene-II are flat. The molecules of pyrene-IV are different and form two kinds of sandwiches (Fig. 11c): sandwich 1 consists of almost flat molecules, whereas sandwich 2 consists of concave molecules. Their curvature is well seen in the [0 13 8] projection (Fig. 12b). Pyrene-V contains only curved molecules, whose curvature increases with pressure and is well seen in the [31 10 1] projection (Fig. 12c). The two concave molecules which formed sandwiches in pyrene-IV can still be recognized in pyrene-V, but they have a much larger offset with respect to each other.

## Exploring intermolecular interactions using Hirshfeld surfaces and fingerprint plots

For visualizing and exploring intermolecular interactions in molecular crystals one uses Hirshfeld surfaces and fingerprint plots. This tool is described in detail in a comprehensive review by McKinnon et al.[20]. The molecular Hirshfeld surface envelops the molecule and defines the volume of space where the promolecule electron density exceeds that of all neighboring molecules[21,22]. The Hirshfeld surface itself is defined by the molecule and the proximity of its nearest neighbors, and hence encodes information about intermolecular interactions.

The fingerprint plot represents in a 2D format two different but useful distance measures, the distances from the internal or external atoms ($d_i$ or $d_e$) to the Hirshfeld surface. Thus, the fingerprint plots are highly sensitive to the immediate environment of the molecule[22]. They are unique for a given molecule in a particular polymorphic form.

We used the CrystalExplorer program[23] to construct Hirshfeld surfaces, mapped with shape index, curvedness, and $d_e$ (for details of various functions of distance and curvature mapped on Hirshfeld surfaces see ref. 22), as well as fingerprint plots for all pyrene polymorphs studied here. They are presented in Figs. 13–15. The fingerprint plots highlighting particular intermolecular contacts and interactions for crystallographically distinct molecules of all polymorphs are shown in Supplementary Figs. 1 and 2. These two figures are described in detail in Supplementary Note 1.

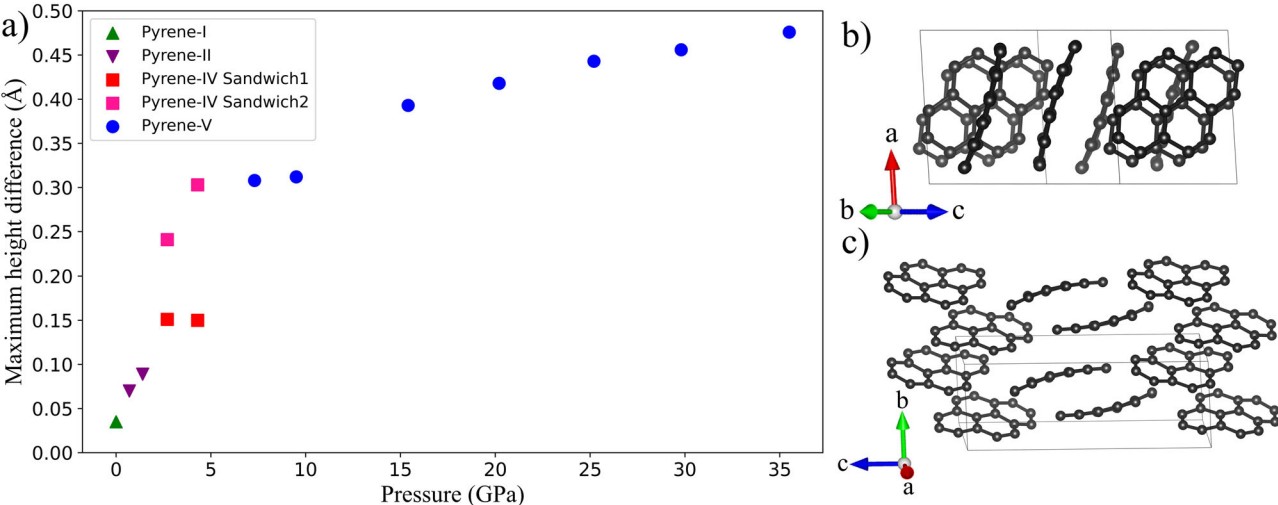

**Fig. 12 | Scatter plot of the maximum surface height difference and crystal structures of pyrene-IV at 4.3 GPa and pyrene-V at 35.5 GPa. a** The pressure dependence of the maximal surface height difference for all studied polymorphs; **b** the structure of pyrene-IV viewed along the [0 13 8] direction; and **c** the structure of pyrene-V viewed along the [31 10 1] direction. C atoms are black, and H atoms are not shown.

**Fig. 13 | Fingerprint plot and Hirshfeld surfaces for pyrene-I at ambient and pyrene-II at 0.7 GPa. a** Crystal-packing diagram in the front and back views of pyrene-I molecule at ambient, with the Hirshfeld surface of the central molecule mapped with shape index; Fingerprint plot and the front and back views of Hirshfeld surface for pyrene-I molecule at ambient (**b**) and pyrene-II molecule at 0.7 GPa (**c**), mapped with shape index, curvedness and $d_e$. The front view depicts the arrangement of the four carbon rings as ABCD, while the back view shows ACBD. Shape index is mapped from −1.0 (red) to 0.0 (green) to 1.0 (blue). Curvedness is mapped from −4.0 (red) to 0.0 (green) to 1.0 (blue). Distance external to the surface, $d_e$, is mapped over the range 1.0 (red) Å to 1.75 (green) Å to 2.5 (blue) Å. In the Fingerprint plot, blue corresponds to the low frequency of occurrence of a ($d_i$, $d_e$) pair, while red points indicate the high frequency of the surface points with that ($d_i$, $d_e$) combination.

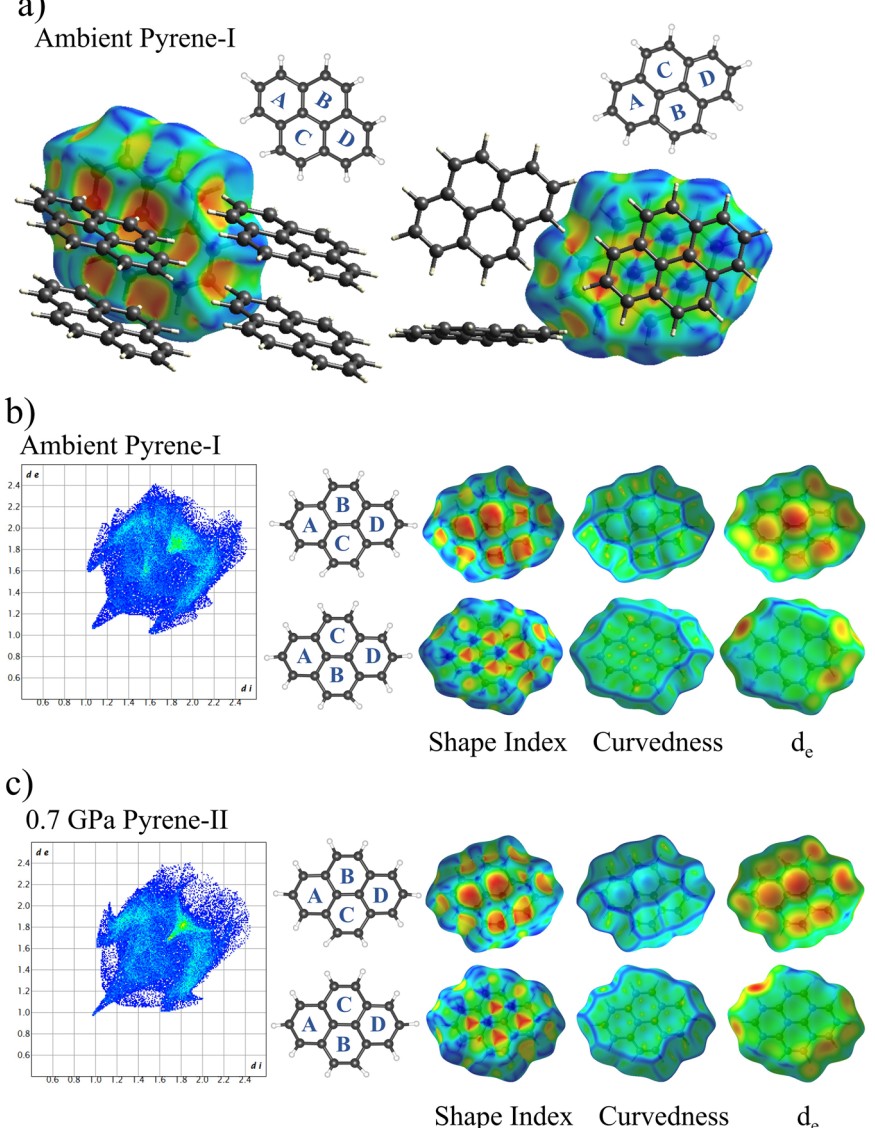

**Fig. 14 | Fingerprint plot and Hirshfeld surfaces for pyrene-IV at 2.7 GPa.** Fingerprint plot and the front and back views of Hirshfeld surface for pyrene-IV sandwich1 molecule (**a**) and pyrene-IV sandwich2 molecule (**b**) at 2.7 GPa, mapped with shape index, curvedness, and $d_e$. Distance external to the surface, $d_e$, is mapped over the range 0.9 (red) Å to 1.6 (green) Å to 2.3 (blue) Å.

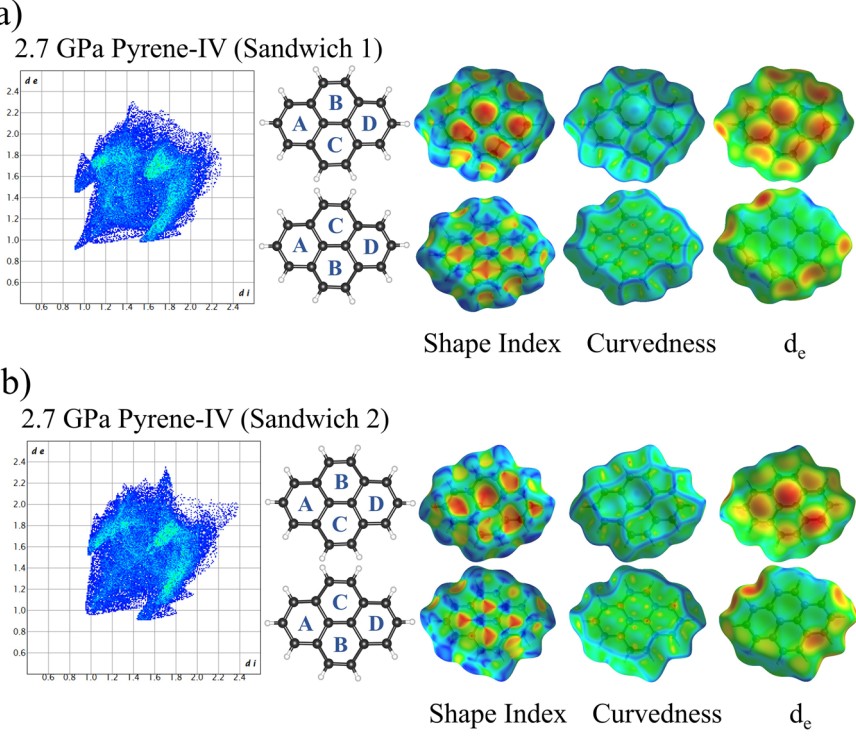

**Fig. 15 | Fingerprint plot and Hirshfeld surfaces for pyrene-V at 7.3 GPa and 35.5 GPa.** Fingerprint plot and the front and back views of Hirshfeld surface for pyrene-V molecule at 7.3 GPa (**a**) and 35.5 GPa (**b**), mapped with shape index, curvedness, and $d_e$. Distance external to the surface, $d_e$, is mapped over the range 0.8 (red) to 1.5 (green) to 2.1 (blue) Å for pyrene-V molecule at 7.3 GPa and from 0.7 (red) Å to 1.3 (green) Å to 1.8 (blue) Å at 35.5 GPa.

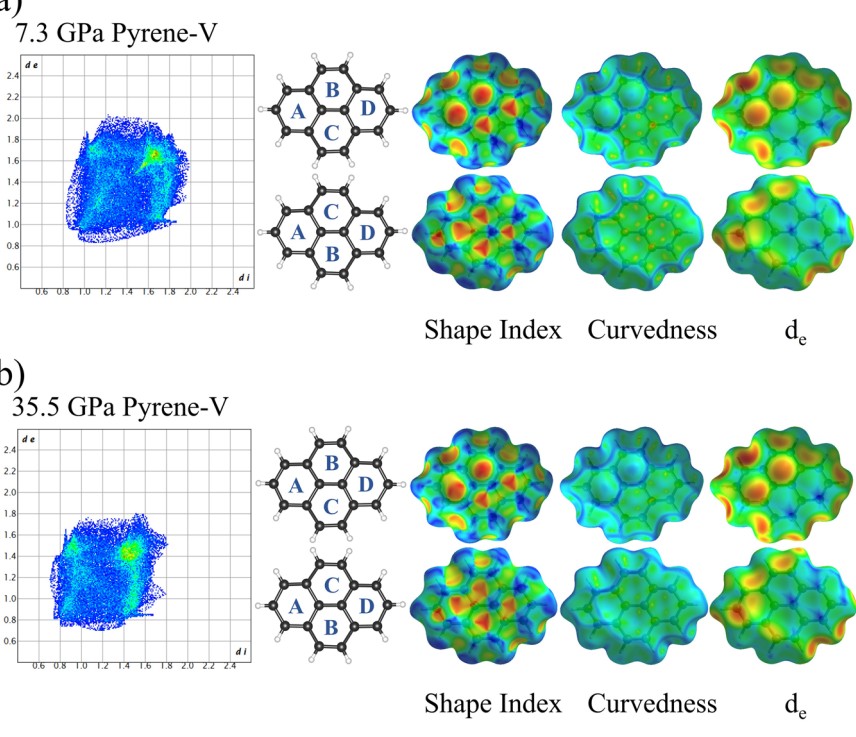

Figure 13a shows the crystal-packing diagram for molecules in pyrene-I. In pyrene-I, considering its sandwich herringbone structure, one side of each molecule in a pair is dominated by π···π stacking interactions (backside, turned to the other molecule in the pair), whereas the other (front) side is dominated by C–H···π contacts. The head-to-head H···H contacts of about 2 Å are very short[20]. The Hirshfeld surfaces of the molecule in pyrene-I, as viewed from the front side (upper row) and back side (bottom row), are shown in Fig. 13b. The alternating red and blue triangles in the back view of the Hirshfeld surface mapped with shape index indicate π···π stacking interactions that are characteristic of graphite-like layered packing[20]. The two molecules in the pair pack in offset. The blue color indicates convex curvatures. The large red-colored regions of conclave curvature on the other side (front view) reflect the C–H···π interactions between pyrene molecules.

The $\pi\cdots\pi$ stacking motif is manifested in the two-dimensional fingerprint plot as a green triangular-shaped region with the minimal $d_e \approx d_i$ at ~1.75 Å (Fig.13b, left), which is consistent with the previously mentioned interplanar distance of ~3.5 Å between the two molecules in a pair (Supplementary Table 15). In pyrene-II at 0.7 GPa (Fig. 13c, left), due to compression, the corresponding $d_e \approx d_i$ point in the fingerprint plot is at ~1.65 Å which implies the decrease of the intermolecular distance in pyrene-II down to ~3.4 Å. The two pairs of wings represent two distinct main C–H$\cdots\pi$ (donor and acceptor) interactions. A sharp feature in the lower left corner of the fingerprint plot is a manifestation of a short head-to-head H$\cdots$H contact. With the pressure increase (Fig. 13c), the two pairs of wings in the fingerprint plot of pyrene-II representing the main C–H$\cdots\pi$ interactions overlap, suggesting them to become less distinct with pressure.

Hirshfeld surfaces and fingerprint plots of pyrene-I at ambient conditions have been previously discussed by McKinnon et al.[20], and those of pyrene-I (with reference to the ambient-pressure structure at 113 K) and pyrene-II (with reference to the ambient-pressure structure at 93 K) by Fabbiani et al.[6]. Our observations are in accordance with the literature data.

Triclinic pyrene-IV possesses two crystallographically distinct molecules. Those in sandwich-1 (Fig. 14a), still almost flat, feature some asymmetry in the fingerprint plot clearly seen in its "wings" (Fig. S2), but preserve head-to-head H$\cdots$H contacts. In sandwich-2 (Fig. 14b), the alternating red and blue triangles in the Hirshfeld surface mapped with shape index indicate graphite-like stacking, whereas that mapped with curvedness shows the decrease of the flat contact area on the back side reflecting a substantial curvature of the molecules. The two main C–H$\cdots\pi$ interactions mapped with $d_e$ show their pronounced difference both in the back and front sides if compared with sandwich-1. In addition, the tip in the lower left corner of the fingerprint plot disappears in sandwich-2, indicating that there is no longer a short head-to-head H$\cdots$H contact between the molecules.

In pyrene-V, the molecules are stacked in a simple herringbone motif that immediately reflects in the fingerprint plot and the Hirshfeld surfaces (Figs. 15 and S2e,f). The $\pi\cdots\pi$ stacking interactions, manifesting as the alternating red and blue triangles in the Hirshfeld surface mapped with shape index[20], appear now on both the back and front sides. Although crystallographically equivalent molecules in pyrene-V are related by an inversion center, they do not form sandwiches. At 35.5 GPa, the size of the area of the fingerprint plot reduces considerably and the wings on both sides almost disappeared, suggesting a high density of molecular stacking (Fig. 15b). The regularity in the distribution of blue and red triangles in both front and back sides of the Hirschfeld surface (shape index) is lost, indicating a considerable departure from graphite-like stacking due to a large curvature of molecules.

For comparison with other polymorphs, we built fingerprint plots of pyrene-III at 0.3 GPa using the structure data from ref. 6 (Fig. S2g), as we did not have our own structural data for pyrene-III of sufficient quality. Interestingly, the fingerprint plot of pyrene-III (Fig. S2g) appeared to be substantially different from those of other polymorphs. However, the absence of head-to-head H$\cdots$H contacts and the large contribution of $\pi\cdots\pi$ interactions in the molecular bonding may be counted as similarities of pyrene-III and pyrene-V, sharing a common herringbone structural motif. A general glance at the evolution of Hirshfeld surfaces and fingerprint plots of metastable pyrene polymorphs allows us to deduce that the change in molecular interactions under pressure is continuous and the dominating interactions are common for polymorphs with similar stacking motifs.

To summarize, our study represents significant progress in high-pressure structural investigations of PAHs using SC-XRD, which were previously limited to very low pressures of around 2 GPa (naphthalene was studied up to 2.1 GPa by Fabbiani et al.[6]. Here, the behavior of pyrene was studied under compression up to 35.5 GPa using synchrotron SC-XRD in a diamond anvil cell with helium as a soft quasi-hydrostatic pressure transmitting medium. Previous structural investigations of pyrene were conducted on single crystals recrystallized from solution in dichloromethane under pressure, which is limited by the freezing pressure of dichloromethane of 1.33 GPa, leading to the deterioration of the single crystal[6]. Higher pressures in structural investigations of PAHs have never been explored before, because it was anticipated that large molecules have low conformational flexibility under pressure and the experience with direct compression of organic compounds with larger molecules showed this method to be ineffective[6].

In our study, it was found that at 0.7 GPa, pyrene-I transforms to metastable pyrene-II, whose structure under high pressure at room temperature is reported here for the first time. It is in good agreement with previous low temperature—ambient pressure data[8–10]. Two other transformations to previously unknown metastable polymorphs, pyrene IV and pyrene V, were observed at 2.7 GPa and 7.3 GPa, respectively. Pyrene-V was preserved in He pressure medium up to ~35 GPa due to fully unexpected structure compaction accompanied by considerable deformation of molecules and their alignment along the crystallographic $c$-axis, enabling avoiding direct H–H contacts. Our experiments reveal that gradual compression results in continuous compaction of molecular packing, eventually leading to curvature of molecules, which has never been observed before under pressure, although twisted organic molecules are known, for example, in diperinaphthyleneanthracene (NAPANT01), whose Hirshfeld surface reflects the significant twist in the molecular structure caused by repulsion between H atoms[20].

Upon compression, the molecule packing motif changed from sandwich-herringbone, distinctive for pyrene-I, pyrene-II, and pyrene-IV, to simple herringbone in pyrene-V. Interestingly, the herringbone motif is characteristic of the structure of both low-pressure pyrene-III and high-pressure pyrene-V, whose molecules are drastically different in shape—they are flat in the former and substantially curved in the latter. Extending the pressure range of structural studies of organic material to over 35 GPa enabled us to demonstrate that the compression of crystals of organic materials in a quasi-hydrostatic medium can lead to the formation of numerous unexpected metastable polymorphs.

To conclude, our experimental study contributes to the fundamental understanding of the polymorphism of PAHs, their behavior under non-ambient conditions, and the evolution of chemical bonding affecting the structure–property relationships of compounds of the important class of organic materials.

Our results highlight the need for a deeper understanding of the observed phases and phenomena using theoretical methods. This is particularly important given the proven potential of high-pressure techniques to alter material properties, as has been successfully demonstrated and exploited in the synthesis of inorganic materials like superhard diamond and cubic boron nitride. Our study indicates that similar potential exists for organic materials. Therefore, further research, including computational studies, aimed at exploring the capabilities of high pressure for synthesizing organic materials with unique properties, is highly warranted and timely.

## Methods

### Sample preparation

A crystalline powder of pyrene ($C_{16}H_{10}$) of 99.9% purity was purchased from Merck. Single crystals were selected under an optical microscope and preselected for high-pressure XRD studies in DAC #1 at ambient pressure (see Supplementary Table 1 for the summary of all experiments). The preselected crystals, averaging about 10 μm in size, and pieces of ruby, approximately 15 μm in size, were loaded in DACs equipped with diamonds with the culets size of 250 μm and a rhenium gasket with a thickness of 40 μm and a hole of ~120 μm in diameter. In all experiments, a gold micrograin was placed inside the center of the pressure chamber along with the sample to facilitate locating the center of rotation. As a pressure-transmitting medium, neon (Ne) or helium (He) was used. The DACs, BX90-type[24] (DAC #2) or membrane-type (DAC #3), were gradually pressurized to 17 GPa and to 35 GPa, respectively.

## Single-crystal XRD

The SC-XRD studies at room temperature were carried out in DAC #1, DAC #2 at the ID15B beamline ($\lambda$ = 0.4100 Å, ESRF) and DAC #3 at the ID27 beamline ($\lambda$ = 0.3738 Å, ESRF). The pressure was determined by the ruby luminescence method[25]. At each pressure step, the data were collected in step scans of 0.5° upon rotating the DAC from −34° to +34° about the vertical axis ($\omega$-scan). For single-crystal data analysis (peak search, unit cell finding, and data integration), the CrysAlisPro Software[26] was employed, whereas the crystal structures were determined using SHELX[27] and refined utilizing the OLEX2 software[28]. The high-pressure XRD data did not allow anisotropic refinement. All refinements were made in isotropic approximation. No twinning was observed. Hydrogen atoms were added using the riding constraint (HFIX instructions) to automatically constrain their positions in OLEX2.

## Theoretical calculations

Structural relaxation, static enthalpy, and eDOS were determined through first-principles calculations employing the Kohn-Sham DFT framework with the generalized gradient approximation as proposed by Perdew–Burke–Ernzerhof[29]. This approximation was integrated within the projector augmented-wave method[30] to describe the exchange and correlation within the Vienna Ab initio Simulation Package (VASP)[31]. Additionally, we employ the DFT-D3 method for dispersion correction[32]. For Brillouin zone sampling, we employed the Monkhorst-Pack scheme[33] with a k-point grid of $2 \times 3 \times 4$ for pyrene-I and pyrene-II, $2 \times 8 \times 4$ for pyrene-III, $4 \times 3 \times 3$ for pyrene-IV and $4 \times 2 \times 2$ for pyrene-V. Furthermore, an energy cutoff of 520 eV was applied to the plane-wave expansion. All structures were relaxed until the energy difference for the electronic self-consistent calculation was smaller than $10^{-5}$ eV/cell and the Hellman−Feynman forces became less than $2 \times 10^{-3}$ eV/Å. Structural optimizations for all considered phases were performed with PBE. In order to accurately describe eDOSs of pyrene polymorphs, we used PBE0[34].

## Data availability

CCDC number refers to the supplementary crystallographic data for this paper. These data can be obtained free of charge from The Cambridge Crystallographic Data Center via www.ccdc.cam.ac.uk/structures.

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

## Acknowledgements

The authors acknowledge the European Synchrotron Radiation Facility (ESRF) for the provision of beamtime at the ID15b and ID27 beamlines. Computations were performed at the Leibniz Supercomputing Center of the Bavarian Academy of Sciences and the Humanities, and the Research Center for Scientific Computing at the University of Bayreuth. N.D. and L.D. thank the Deutsche Forschungsgemeinschaft (DFG projects DU 945/15-1; LA 4916/1-1; DU 393–9/2, DU 393–13/1) for financial support. N.D. also thanks the Swedish Government Strategic Research Area in Materials Science on Functional Materials at Linköping University (Faculty Grant SFO-Mat-LiU No. 2009 00971). D.L. thanks the UKRI Future Leaders Fellowship (MR/V025724/1) for financial support. W.Z. thanks L. Man for the helpful discussions. For the purpose of open access, the author has applied a Creative Commons Attribution (CC BY) license to any Author Accepted Manuscript version arising from this submission.

## Author contributions

L.D. and N.D. conceived the overall project. W.Z. prepared the high-pressure experiments. W.Z., Y.Y., D.L., A.A., E.B., A.P., M.H., T.P., and M.M. performed the synchrotron X-ray diffraction experiments. W.Z. performed the synchrotron X-ray diffraction data analysis and the theoretical calculations. W.Z., L.D., and N.D. analyzed all the data. W.Z., L.D., and N.D. wrote the manuscript with input from all the other authors. All the authors discussed and contributed to the manuscript.

## Funding

## Competing interests

The authors declare no competing interests.
