## [Peer Review File · Communications Chemistry]

Reviewers' comments:

Reviewer #1 (Remarks to the Author):

The authors present experiments and accompanying calculations on various polymorphs of pyrene under extreme pressures, with the experiments performed using diamond anvil cells. Though it's not quite my area of expertise, the experiments all appear sensible and form, in my view, the core of the value of this paper.

There's nothing wrong at all with the fundamentals of this paper, but the accompanying analysis is disappointing in that it has the beginnings of very interesting work, but the authors appear to have stopped short of digging into details and finding broader scientific points for discussion. After reading this work I'm at a bit of a loss as to what was learned or established scientifically by performing these experiments - i.e. what is the takeaway message? That's not meant to be critical of performing the work, but the presentation and discussion are not really showcasing the value of the work.

The calculations and accompanying analysis feels a bit like an afterthought. There is a section devoted to geometrical analysis of the polymorphs which is the most interesting aspect of the analysis presented, but even here the authors do not proceed to associate energetic details in intermolecular interactions or intramolecular strain with their geometric information. The authors clearly had the resources to perform such calculations (even just single point calculations on molecular pairs at the fixed geometries) as they used VASP to optimise the various crystal structures at different pressures and calculated enthalpies, but the link between the theoretical calculations and the broader analysis is not really made.

No clear association is made between this work on pyrene and the (cited) previous work on pyrene at lower pressures - does this work have the same broad trends established elsewhere, does it contradict anything?

The calculation of Hirshfeld surfaces and associated fingerprints makes sense and can be useful as a visual guide to rationalise where the structure and interactions change. However, it is my view that too much space is devoted to this discussion - much of which is not particularly illuminating when compared with the possibility of actually computing or extracting energetic information. To this end, DFT calculations have been performed at a sensible level of theory, so it leaves the question as to why e.g. the energetic strain in the molecule at the various crystal pressures was not evaluated? Such information would undoubtedly be more useful (and could be discussed with regard to the experimental strain curves)

Something missing that would significantly improve this work is discussion of vibrational thermal contributions to free energy differences between polymorphs, and how this affects the relative energies of the polymorphs at different pressures.

Very minor point:

Theoretical section details: The details of the PBE (plane-wave DFT) calculations are great, verging on too much information. By contrast, the sentence "In order to accurately describe the³⁵⁸ electronic properties of pyrene polymorphs, we used the PBE0" is not enough detail to reproduce or understand what was done. This section should be made clear - were optimisations performed with PBE0 as well, or just PBE? Which systems were calculated using PBE/PBE0 etc. Just more detail is needed here.

Reviewer #2 (Remarks to the Author):

The paper by Zhou et al. investigates the compression behavior of pyrene polymorphs up to 35.5 GPa using synchrotron single-crystal X-ray diffraction. It identifies novel metastable polymorphs and observes structural changes during compression. The study employs a combination of experimental and theoretical approaches to analyze crystal structures, compressional behavior, theoretical calculations, geometrical properties, and intermolecular interactions. Key findings include structural transitions, molecular curvature under pressure, and insights into molecular interactions. The paper contributes to understanding polymorphism in organic materials but requires improvements in clarity, organization, and contextualization. Overall, while the paper addresses an interesting scientific question, it may need to better emphasize the significance of its experimental achievements and provide clearer explanations of its methodologies and findings.

Below are more detailed comments for each section:

Introduction:

- The introduction provides a broad overview of the importance of PAHs but lacks clarity regarding the specific focus of the study.
- It would be beneficial to clearly articulate the objectives and significance of the research within the introduction to provide better context for readers.
- While previous research is mentioned, a more thorough discussion of how it leads to the current investigation and its relevance would enhance the introduction's effectiveness.

- The introduction could be streamlined by focusing on key points and providing a smoother transition to the subsequent sections.

Crystal structures of pyrene polymorphs:

- The section offers valuable insights into the structural transitions of pyrene under pressure but could benefit from improved organization and clarity.
- Providing a clearer flow of information and summarizing key parameters directly in the text would enhance readability.
- A more comprehensive comparison with existing literature would provide a clearer context for the significance of the current findings.

Compressional behavior of the polymorphs of pyrene:

- Summarizing key findings or trends could help distill the most important information and prevent overwhelming readers with data.
- Some explanations of lattice parameter variations could be further clarified, particularly for readers less familiar with crystallography.

Theoretical Calculations.

- The section offers valuable insights into the behavior of pyrene polymorphs under pressure but could address discrepancies between calculated and experimental data more thoroughly.
- Further discussion on the significance of the thermodynamic stability analysis and the implications for understanding pyrene polymorph behavior under pressure would enrich the section.
- Expanding on the implications of the observed color change in pyrene crystals and its connection to electronic properties would deepen the analysis.

Geometrical analysis of the structures of the pyrene polymorphs:

- The section provides a thorough examination of the geometric properties of pyrene polymorphs but could improve accessibility by providing additional context for technical terms and concepts.
- Integrating experimental observations with theoretical analysis enhances the discussion and validates the findings.
- The presentation of figures and tables effectively communicates the observed data and interpretations, contributing significantly to our understanding of structural dynamics in organic solids.

Exploring intermolecular interactions using Hirshfeld surfaces and fingerprint plots:

- Comparing the findings with previous studies and discussing potential applications would provide a broader context for the research.

Summary

- The summary effectively outlines the main findings and significance of the study but could provide more specific details about the experimental methodology and quantitative results.
- Comparing the findings with previous studies and discussing potential applications would enhance clarity and depth.
- Addressing these aspects would strengthen the summary's comprehensiveness and critical analysis of the study.

Addressing these points would significantly improve the overall quality and impact of the paper, making it more valuable to the scientific community.

Moreover, expanding the study to include crystal lattice energy computation using software like CRYSTAL or CLP-PIXEL could provide valuable insights into the energetics of the pyrene polymorphs under pressure. By calculating the crystal lattice energy, researchers can gain a deeper understanding of the stability and relative energetics of different polymorphs, which can further elucidate their structural transitions and behavior under compression. Comparing the computed crystal lattice energies with experimental data and theoretical calculations can provide a more comprehensive understanding of the structural transitions observed under pressure. This comparative analysis can highlight discrepancies between different methodologies and shed light on the underlying factors influencing the stability of the pyrene polymorphs.

Additionally, the information about structural refinement is poorly reported – please provide detailed information about isotropic/anisotropic refinement, hydrogen treatment, twinning, etc.

Reviewer #3 (Remarks to the Author):

The authors reported the phase transition of pyrene under HP by single crystal XRD. The data look good and are valuable. I would like to recommend the paper to be published after the following improvement.

1. The literature and evidence of the phase transitions should be clearly presented and discussed. For example, the references, like [5] and [7] both claimed phase transition, but to different phases. Could the author make an explanation based on their investigation? The current research should be connected to the previous researches.
2. The phase I and II have very similar structures. The authors need to give an explanation why they are two phases. Currently there are not enough data points to conclude a non-continuous EOS, then an energy barrier should be found by theoretical investigation, or theoretical calculation of Raman spectra should be compared to the literature.
3. Similar illustration based on energy barrier should be done for other transitions.

4. The local structure of the stacking should be discussed in details to illustrate the difference between these phases. For example, is the carbon located on the center of the ring of the next molecule, or the bond center, etc.? What is the geometry of inter-sandwich interaction?-Phase V is still a stacking of sandwich. No need to move it out of group.

5. It is a little bit weird of using KCl as PTM to prepare phase III, because the bulk modules of KCl is much bigger than pyrene. Does the author want to make a non-hydrostatic condition? The obtained powder XRD pattern can not support the identification of phase III due to the preferred orientation. I would suggest the author the remove the experimental investigation of phase III from this paper, and only leave the theoretical discussion of phase III for comparison. The energy barrier to phase III should also be calculated from phase I, II, IV and V, which should be bigger and shows why the author did not observe phase III in their experiment.

Reply to Reviewers' comments

Reviewer #1 (Remarks to the Author):

The authors present experiments and accompanying calculations on various polymorphs of pyrene under extreme pressures, with the experiments performed using diamond anvil cells. Though it's not quite my area of expertise, the experiments all appear sensible and form, in my view, the core of the value of this paper.

There's nothing wrong at all with the fundamentals of this paper, but the accompanying analysis is disappointing in that it has the beginnings of very interesting work, but the authors appear to have stopped short of digging into details and finding broader scientific points for discussion. After reading this work I'm at a bit of a loss as to what was learned or established scientifically by performing these experiments - i.e. what is the takeaway message? That's not meant to be critical of performing the work, but the presentation and discussion are not really showcasing the value of the work.

We highly appreciate Reviewer's #1 positive view on the experiments and the fundamental aspects of the paper. Regarding his/her concern, we have to admit that we likely insufficiently elaborated the details and the takeaway message for a broader readership of the *Communications Chemistry*. (For us the impact of the work was self-clear.) We thank Reviewer #1 for pointing this out.

Following Reviewer's #1 suggestion, we have extended the discussions and explicitly formulated the major impact of the research in the field and beyond. Appropriate amendments have been done in the text. Abstract, introduction, and summary have been re-worked substantially. In particular, we have explained that our study represents a considerable progress in high-pressure structural investigations of polycyclic aromatic hydrocarbons (PAHs), which were previously limited by very low pressures of around 2 GPa; the structure solution for pyrene-II at high pressure is reported in our work for the first time. Contrary to the previous belief that large organic molecules have low conformational flexibility under pressure, we have demonstrated that gradual compression results in continuous compaction of molecular packing, eventually leading to curvature of molecules, which has never been observed before under pressure. Our results revealed that pyrene-V can be preserved in He pressure medium up to ~35 GPa due to fully unexpected structure compaction accompanied by considerable deformation of molecules and their strong alignment along one crystallographic axis.

All modifications introduced to the text and figure captions to provide clearer explanations of our methodologies and findings are highlighted in yellow in the revised manuscript. Those parts which left unchanged, but moved to other parts of the manuscript for better flow, have not been marked.

The calculations and accompanying analysis feels a bit like an afterthought. There is a section devoted to geometrical analysis of the polymorphs which is the most interesting aspect of the analysis presented, but even here the authors do not proceed to associate energetic details in intermolecular interactions or intramolecular strain with their geometric information. The authors clearly had the resources to perform such calculations (even just single point calculations on molecular pairs at the fixed geometries) as they used VASP to optimise the

various crystal structures at different pressures and calculated enthalpies, but the link between the theoretical calculations and the broader analysis is not really made.

We agree with Reviewer #1, that it would be an interesting task to proceed with the suggested calculations, but they are out of scope of the experimental study we present in this manuscript. We would like to emphasize that in this experimental work theory is used in order to confirm that (a) the structural changes we observe are not arbitrary, and (b) that theory indeed supports the experimental findings. Whatever additional calculations could be made, they will not affect the results of our experimental study.

No clear association is made between this work on pyrene and the (cited) previous work on pyrene at lower pressures - does this work have the same broad trends established elsewhere, does it contradict anything?

It would be nice to compare trends, but the point is that structural investigations using single-crystal XRD, and only up to a very low pressure of 0.5 GPa, have been made only by Fabbiani et al (2006). Other papers report the results of vibrational spectroscopy studies, which cannot provide any explicit structural information. Surely, we have compared our results with those of Fabbiani et al., which were reflected both in the text and Figures 2 and 4 of the original manuscript. As written in the paper: "In our room temperature experiments, we did not observe pyrene-III described in the study of Fabbiani et al. [6], where it was synthesized through the recrystallisation from a 0.5 M solution of pyrene in dichloromethane after several temperature-annealing cycles (slow cooling and heating between 303 K and 293 K) at 0.3 GPa. This work and our computational result motivated us to conduct a high-pressure high-temperature experiment..." This HPHT experiment revealed formation of pyrene-III after heating the sample at 4 GPa and 473 K. As we have written in the paper, "This observation suggests that pyrene-III is likely a thermodynamically stable phase at pressures above 0.3 GPa, whose synthesis requires heating to overcome the energy barrier". High-pressure structural data on pyrene-II have been absent in the literature, as pyrene-II was previously reported only at low temperature and ambient pressure. We did observe pyrene-II at 0.7 and 1.4 GPa for the first time. A broader discussion of the previous structural studies of pyrene-II, as well as spectroscopic evidences of a phase transition has been added in Introduction (marked in yellow).

The calculation of Hirshfeld surfaces and associated fingerprints makes sense and can be useful as a visual guide to rationalise where the structure and interactions change. However, it is my view that too much space is devoted to this discussion - much of which is not particularly illuminating when compared with the possibility of actually computing or extracting energetic information. To this end, DFT calculations have been performed at a sensible level of theory, so it leaves the question as to why e.g. the energetic strain in the molecule at the various crystal pressures was not evaluated? Such information would undoubtedly be more useful (and could be discussed with regard to the experimental strain curves)

We agree with the Reviewer #1 that "The calculation of Hirshfeld surfaces and associated fingerprints makes sense and can be useful as a visual guide to rationalise where the structure and interactions change". This is exactly why we have done this following the common practice in the field of research on molecular compounds; for these Hirshfeld surfaces calculations we did not need any other information except our own experimental data. Further theoretical analysis, which is out of scope of the present manuscript, may be of interest of other researches specialised on theoretical calculations, for whom our experimental findings (crystal structures of two new pyrene polymorphs, IV and V) have supplied an excellent

starting point. We have emphasised this in the summary of the revised manuscript as follows: “Our results highlight the need for a deeper understanding of the observed phases and phenomena using theoretical methods. This is particularly important given the proven potential of high-pressure techniques to alter material properties, as has been successfully demonstrated and exploited in the synthesis of inorganic materials like superhard diamond and cubic boron nitride. Our study indicates that similar potential exists for organic materials. Therefore, further research, including computational studies, aimed at exploring the capabilities of high pressure for synthesizing organic materials with unique properties, is highly warranted and timely.”

Something missing that would significantly improve this work is discussion of vibrational thermal contributions to free energy differences between polymorphs, and how this affects the relative energies of the polymorphs at different pressures.

As we have pointed out before, theory in this experimental work serves to support our findings and conclusions. We have made the enthalpy differences (ΔH) calculations at 0K for the four polymorphs (pyrene-II, pyrene-III, pyrene-IV, and pyrene-V) relative to pyrene-I (Table S14, Fig. 5) to elucidate their relative stability. They revealed that up to ca. 2 GPa, pyrene-II is relatively more stable than pyrene-IV and pyrene-V, whereas above this pressure, pyrene-V is relatively more stable). Pyrene-III appears to be the thermodynamically stable phase if compared to all other polymorphs above 0.03 GPa and up to 5 GPa. The discussion of vibrational thermal contributions is a subject of separate research for theory specialists.

Very minor point:

Theoretical section details: The details of the PBE (plane-wave DFT) calculations are great, verging on too much information. By contrast, the sentence "In order to accurately describe the electronic properties of pyrene polymorphs, we used the PBE0" is not enough detail to reproduce or understand what was done. This section should be made clear - were optimisations performed with PBE0 as well, or just PBE? Which systems were calculated using PBE/PBE0 etc. Just more detail is needed here.

We thank the Reviewer for this comment. Following the Reviewer's #1 request, we have added appropriate information in the Theoretical section: Structural optimisations for all considered phases were performed with PBE. In order to accurately describe eDOSs of pyrene polymorphs, we used PBE0.

Reviewer #2 (Remarks to the Author):

The paper by Zhou et al. investigates the compression behavior of pyrene polymorphs up to 35.5 GPa using synchrotron single-crystal X-ray diffraction. It identifies novel metastable polymorphs and observes structural changes during compression. The study employs a combination of experimental and theoretical approaches to analyze crystal structures, compressional behavior, theoretical calculations, geometrical properties, and intermolecular interactions. Key findings include structural transitions, molecular curvature under pressure, and insights into molecular interactions. The paper contributes to understanding polymorphism in organic materials but requires improvements in clarity, organization, and contextualization. Overall, while the paper addresses an interesting scientific question, it may need to better emphasize the significance of its experimental achievements and provide clearer explanations of its methodologies and findings.

Below are more detailed comments for each section:

We thank Reviewer #2 for positive evaluation of our work we are very grateful for all his/her constructive suggestions aimed at improvements in clarity, organization, and contextualization of the paper. We have done our best to follow Reviewer's #2 recommendations while revising our manuscript. As the comments for each section below contain both praises and criticism, we thank the Reviewer for all positive words and address the concerns in our reply below.

Introduction:

- The introduction provides a broad overview of the importance of PAHs but lacks clarity regarding the specific focus of the study.
- It would be beneficial to clearly articulate the objectives and significance of the research within the introduction to provide better context for readers.
- While previous research is mentioned, a more thorough discussion of how it leads to the current investigation and its relevance would enhance the introduction's effectiveness.
- The introduction could be streamlined by focusing on key points and providing a smoother transition to the subsequent sections.

The Introduction was re-worked considering all Reviewer's #2 suggestions. In particular, the specific focus of the study was clarified, the objectives and significance of the research have been explicitly articulated, the discussion of previous research was enlarged, and the issues previously unresolved have been explained. All these is highlighted in yellow in the revised manuscript.

Crystal structures of pyrene polymorphs:

- The section offers valuable insights into the structural transitions of pyrene under pressure but could benefit from improved organization and clarity.
- Providing a clearer flow of information and summarizing key parameters directly in the text would enhance readability.
- A more comprehensive comparison with existing literature would provide a clearer context for the significance of the current findings.

This part contains the description of the crystal structures. The key crystallographic and experimental parameters are summarised in Tables and Figures and, if relevant (for pyrene-I and pyrene-II), a comparison with the previous data is provided in the same tables. On Reviewer's request, we have expended the discussion of previous data:

“A transition from pyrene-I to pyrene-II below 110 K was first reported by Jones et al. (1978) [8], and the structure of pyrene-II was suggested on the basis of a combination of micro-electron diffraction and atom—atom, pairwise potential calculations. Knight et al. (1996) [9] confirmed and refined the pyrene-II structure from high-resolution neutron powder diffraction data collected from a fully deuterated sample at 4.2 K. First single-crystal XRD analysis of pyrene-II at 93 K and ambient pressure was reported by Frampton et al. (2000) [10]. Our work reports the first structural analysis of pyrene-II under pressure at room temperature using single-crystal XRD and provides crystallographic data for pyrene-II, which are in a very good agreement with those obtained at low temperature and ambient pressure [8-10].”

Compressional behavior of the polymorphs of pyrene:

- Summarizing key findings or trends could help distill the most important information and prevent overwhelming readers with data.
- Some explanations of lattice parameter variations could be further clarified, particularly for readers less familiar with crystallography.

A short summary concerning compressional behaviour of the polymorphs have been added at the end of this section. Additional explanations to the Figure 4 caption (on lattice parameter variations) have been added for better clarity.

Theoretical Calculations.

- The section offers valuable insights into the behavior of pyrene polymorphs under pressure but could address discrepancies between calculated and experimental data more thoroughly.
- Further discussion on the significance of the thermodynamic stability analysis and the implications for understanding pyrene polymorph behavior under pressure would enrich the section.
- Expanding on the implications of the observed color change in pyrene crystals and its connection to electronic properties would deepen the analysis.

The discrepancies between calculated and experimental data in our work are within usual discrepancies characteristic for the method (in our calculations the differences in lattice parameters are of ~3% or less; those up to 5% are considered as reasonable). The agreement of experiment and theory may be improved by calculations at 300 K, but this is out of our expertise and out of the scope of the current work.

We have made the enthalpy differences (ΔH) calculations at 0K for the four polymorphs (pyrene-II, pyrene-III, pyrene-IV, and pyrene-V) relative to pyrene-I (Table S14, Fig. 5) to elucidate their relative stability just to compare with our observations. On Reviewers' request, we have added the following discussion to the text:

“The calculations suggest that up to 2.07 GPa pyrene-II is relatively more stable than pyrene-IV and pyrene-V. We observed its formation at 0.7 and 1.4 GPa in our room temperature experiment that agrees with the calculations. Above 2.07 GPa pyrene-V is predicted to be more stable than other polymorphs except pyrene-III. This does not contradict to our observations, as we detected pyrene-V formation at 7.3 GPa at the pressure step from 4.3 GPa. It is known that formation of metastable phases is very sensitive to many parameters like stress, for example, which cannot be fully controlled in a DAC experiment, as we have shown previously in our work on high pressure phases of silica [16]”.

Concerning color change, we needed its explanation and it was found. As written in the paper, “Calculations of the electronic density of states (eDOS) for pyrene-I at 1 bar and 0 K and pyrene-V at 50 GPa and 0 K (Fig. 8) have shown that the band gap decreases from 3.3 eV to 0.9 eV. This explains the observed color change of the crystals”. We do not see any basis for further discussions or implications.

Geometrical analysis of the structures of the pyrene polymorphs:

- The section provides a thorough examination of the geometric properties of pyrene polymorphs but could improve accessibility by providing additional context for technical terms and concepts.
- Integrating experimental observations with theoretical analysis enhances the discussion and validates the findings.
- The presentation of figures and tables effectively communicates the observed data and interpretations, contributing significantly to our understanding of structural dynamics in organic solids.

We thank the Reviewer for his/her positive feedback.

Exploring intermolecular interactions using Hirshfeld surfaces and fingerprint plots:

- Comparing the findings with previous studies and discussing potential applications would provide a broader context for the research.

Hirshfeld surfaces and fingerprint plots of pyrene-I at ambient conditions have been

previously discussed by McKinnon et al. (2004), and those of pyren-I (with reference to the ambient-pressure structure at 113 K) and pyrene-II (with reference to the ambient-pressure structure at 93 K) - by Fabbiani et al (2006). We saw no differences in our findings. We referred to these works in the paper, but for clarity, we have added this information explicitly in the revised manuscript.

We had written in the manuscript that we built fingerprint plots of pyrene-III at 0.3 GPa using the structure data from Fabbiani et al (2006) for comparison with those of other polymorphs. We had no own structural data for pyrene-III of sufficient quality. This was mentioned in Supplementary Information (in the caption of Figure S2), but now we have added this information to the text of the revised manuscript for clarity.

Pyrene-IV and -V have never been observed before, so that any previous data are absent.

Summary

- The summary effectively outlines the main findings and significance of the study but could provide more specific details about the experimental methodology and quantitative results.
- Comparing the findings with previous studies and discussing potential applications would enhance clarity and depth.
- Addressing these aspects would strengthen the summary's comprehensiveness and critical analysis of the study.

We have enriched the summary with the requested information. Please see the text highlighted in yellow in the revised manuscript.

Addressing these points would significantly improve the overall quality and impact of the paper, making it more valuable to the scientific community.

Moreover, expanding the study to include crystal lattice energy computation using software like CRYSTAL or CLP-PIXEL could provide valuable insights into the energetics of the pyrene polymorphs under pressure. By calculating the crystal lattice energy, researchers can gain a deeper understanding of the stability and relative energetics of different polymorphs, which can further elucidate their structural transitions and behavior under compression. Comparing the computed crystal lattice energies with experimental data and theoretical calculations can provide a more comprehensive understanding of the structural transitions observed under pressure. This comparative analysis can highlight discrepancies between different methodologies and shed light on the underlying factors influencing the stability of the pyrene polymorphs.

We thank the Reviewer #2 again for valuable suggestions for further computational studies. We are confident that as soon as our experimental study is published, many theoreticians will be happy to address all of these questions and conduct theoretical computations suggested by the Reviewer. To clarify this point we have added the following statement to the summary of the research: “To conclude, our experimental study contributes to the fundamental understanding of the polymorphism of polycyclic aromatic hydrocarbons, their behavior under none-ambient conditions, and the evolution of chemical bonding affecting the structure-property relationships of compounds of the important class of organic materials.

Our results highlight the need for a deeper understanding of the observed phases and phenomena using theoretical methods. This is particularly important given the proven potential of high-pressure techniques to alter material properties, as has been successfully demonstrated and exploited in the synthesis of inorganic materials like superhard diamond and cubic boron nitride. Our study indicates that similar potential exists for organic materials. Therefore, further research, including computational studies, aimed at exploring the capabilities of high pressure for synthesizing organic materials with unique properties, is highly warranted and timely.”

Additionally, the information about structural refinement is poorly reported – please provide detailed information about isotropic/anisotropic refinement, hydrogen treatment, twinning, etc.

The high-pressure XRD data did not allow anisotropic refinement. All refinements were made in isotropic approximation. No twinning was observed. Hydrogen atoms were added using the riding constraint (HFIX instructions) to automatically constrain their positions in OLEX2. We have added this information in the “Single-crystal XRD” section in Methods.

Reviewer #3 (Remarks to the Author):

The authors reported the phase transition of pyrene under HP by single crystal XRD. The data look good and are valuable. I would like to recommend the paper to be published after the following improvement.

1. The literature and evidence of the phase transitions should be clearly presented and discussed. For example, the references, like [5] and [7] both claimed phase transition, but to different phases. Could the author make an explanation based on their investigation? The current research should be connected to the previous researches.

Following the Reviewer’s #3 request, we have extended the discussion of the evidence of the phase transitions reported in the literature. We have added the following text in the section “Crystal structures of pyrene polymorphs”:

“A transition from pyrene-I to pyrene-II below 110 K was first reported by Jones et al. (1978) [8], and the structure of pyrene-II was suggested on the basis of a combination of micro-electron diffraction and atom—atom, pairwise potential calculations. Knight et al. (1996) [9] confirmed and refined the pyrene-II structure from high-resolution neutron powder diffraction data collected from a fully deuterated sample at 4.2 K. First single-crystal XRD analysis of pyrene-II at 93 K and ambient pressure was reported by Frampton et al. (2000) [10]. Our work reports the first structural analysis of pyrene-II under pressure at room temperature using single-crystal XRD and provides crystallographic data for pyrene-II, which are in a very good agreement with those obtained at low temperature and ambient pressure [8-10]”.

For the first time pyrene-II was synthesized under pressure and characterized using single-crystal XRD only in our work. Fabbiani et al. (2006) did not observe pyrene-II, but synthesized pyrene-III from a solution at 0.3 and 0.5 GPa, as we discussed in detail in the original version of the manuscript.

We would like to emphasize that there are no structural data in [5] and [7] (Refs. 11 and 12 in the revised manuscript), and the both claims of phase transitions are solely based on the observed changes in the Raman spectra, so that the authors could only judge that some transformations take place, but what exactly happens with the pyrene’s crystal structure remained unknown. This is underlined by the authors themselves: in [7] (new Ref. 11) (Zallen et al. 1976), for example, one reads: “The solid state transition from the normal dimer-structure form (pyrene-I) to the unknown high-density form (pyrene II) is manifested in the abrupt transformation of the phonon spectrum... The structure of pyrene-II is unknown...” This does not provide much space for comparison, but we added more details about spectroscopic evidences of phase transitions in pyrene to the Introduction section:

“Previous structural studies of pyrene, using diffraction methods, enabled to establish its two polymorphs. The first polymorph, pyrene-II (Fig. 1b), was identified upon a transition from pyrene-I at low temperature [8-10]. Its structural motif is similar to that of pyrene-I. The other polymorph, pyrene III (Fig. 1c) was identified on single crystals of pyrene recrystallized from a dichloromethane solution at 0.3 and 0.5 GPa [6]. It was found to have a different molecular packing model and different intermolecular interactions. The structures will be discussed in detail below in relation to our findings.

Although spectroscopic data do not provide explicit information about the structure of solid matter, it is worth noticing that vibrational spectroscopy investigation of pyrene up to about 1 GPa, pointing towards the existence of phase transformations in pyrene under pressure, was made as early as in 1976 [11], when a transition was detected on an abrupt change of the Raman spectrum at ca. 0.4 GPa. Later Raman spectroscopy study [12] detected a transformation at 0.3 GPa on a crystal grown from a dichloromethane solution, and on a crystal pressurized in argon up to 0.6 GPa, interpreted in the both cases as observation of pyrene-III, similar to that described by Fabbiani et al. (2006) [6].”.

2. The phase I and II have very similar structures. The authors need to give an explanation why they are two phases. Currently there are not enough data points to conclude a non-continuous EOS, then an energy barrier should be found by theoretical investigation, or theoretical calculation of Raman spectra should be compared to the literature.

3. Similar illustration based on energy barrier should be done for other transitions.

The phase transitions in our work were recognised solely on the basis of the observed abrupt structural changes, which were detected by single-crystal XRD that is the most reliable way to study structural phase transformations. Indeed, we could describe the P-V behaviour for all pyrene polymorphs by a common continuous EOS, but in fact, the volume may change continuously with pressure, while other structural parameters abruptly change manifesting a solid state transition. This happens, for example, if molecules are tilted with respect to the unit cell axes that is exactly the case for the pyrene-I–pyrene-II transition. To clarify this point we have added the following in the revised manuscript:

...” The structures of the two polymorphs, pyrene-I and pyrene-II (Fig. 1a,b), are very similar. As underlined in previous studies [8], [10], “a small rotation of molecules around the *c*-axis [it corresponds to the *a*-axis in the standard setting $P2_1/c$ used in our paper for space group #14] of the pyrene-I unit cell generates a new structure that is very close in terms of cell dimensions and packing motif to pyrene-II” (cited from Frampton et al. (2000) [10]). Namely this rotation is responsible for considerable change in the molecules interplanar angle (see the analysis below)”.

4. The local structure of the stacking should be discussed in details to illustrate the difference between these phases. For example, is the carbon located on the center of the ring of the next molecule, or the bond center, etc.? What is the geometry of inter-sandwich interaction?-Phase V is still a stacking of sandwich. No need to move it out of group.

We would like to argue regarding this point. To illustrate the differences in stacking we have already built the Hirschfeld surfaces and fingerprint plots. They visualise the details of interactions and the geometry, along with the crystallographic information files (cifs) we provide for all polymorphs at each pressure point (the files are also deposited at The Cambridge Crystallographic Data Centre (CCSD) database and their deposition numbers are given in corresponding tables in Supplementary Information of the revised manuscript). Namely the analysis of the Hirschfeld surfaces and fingerprint plots of pyrene-V gives

evidence that this polymorph does not possess sandwich structure anymore. As we wrote in the paper, “In pyrene-V, the molecules are stacked in a simple herringbone motif that immediately reflects in the fingerprint plot (Figs. 15a, S2) and the Hirshfeld surfaces. As the $\pi \cdots \pi$ stacking interactions dominate now on both back and front sides, there is a general increase of the contribution of C-C interactions in pyrene-V....” Additionally, the alignment of the molecules at a very low angle, their deformation (the molecules are not flat anymore) and their mutual considerable offset in the projection along the *a*-axis suggest collapse of the sandwich packing. So, we do hope that Reviewer #3 agrees with our arguments. For better clarity of the availability of cifs, the following information has been added in Supplementary Information (see footnote of Table S2), as recommended by the CCSD: “CCDC number contains the supplementary crystallographic data for this paper. These data can be obtained free of charge from The Cambridge Crystallographic Data Centre via www.ccdc.cam.ac.uk/structures”.

5. It is a little bit weird of using KCl as PTM to prepare phase III, because the bulk modulus of KCl is much bigger than pyrene. Does the author want to make a non-hydrostatic condition? The obtained powder XRD pattern can not support the identification of phase III due to the preferred orientation. I would suggest the author to remove the experimental investigation of phase III from this paper, and only leave the theoretical discussion of phase III for comparison. The energy barrier to phase III should also be calculated from phase I, II, IV and V, which should be bigger and shows why the author did not observe phase III in their experiment.

We thank Reviewer #3 for the comment and would like to clarify some points. Surely, quasi hydrostatic conditions (using inert gases) would be preferable, but considering the necessity of heating at such low pressures, it is infeasible, as one cannot control the behaviour of gases, which may be even in a supercritical state upon heating at low pressures. Regarding the identification of phase III, we would like to attract attention of Reviewer #3 to the fact that we indexed pyrene-III on the basis of single-crystal XRD data (see Figure 6a), rather than on the powder XRD pattern. In Figure 6b we showed an integrated XRD pattern to better visualise which phases were present in the sample after heating. Such a consideration fully supports our conclusion that the synthesis of pyrene-III likely requires heating to overcome the energy barrier. We did not intend to make any quantitative proposals.

Reviewers' comments:

Reviewer #1 (Remarks to the Author):

The authors have adequately addressed the referees comments, making the necessary changes throughout the work and appropriately justifying where changes/additions were not made.

The purpose and value of the work is now much clearer, and will hopefully be appreciated by the broader crystallographic community.

Reviewer #2 (Remarks to the Author):

I am pleased to report that the authors have successfully addressed all the concerns raised in my initial review. The corrections have significantly improved the quality and clarity of the manuscript. Therefore, I am recommending this revised version for publication.

Reviewer #3 (Remarks to the Author):

The authors made explanations but did not fully answer my questions, and we did not get new knowledge about the principle/rule of the phase transition of aromatics under high pressure, except for the crystal data.

1. The phase transitions and phase boundaries. The authors need to present some extra experimental evidences to support the phase transitions, or make a calculation on the energy barrier of the phase transitions.
2. The authors wrote "Namely the analysis of the Hirschfeld surfaces and fingerprint plots of pyrene-V gives evidence that this polymorph does not possess sandwich structure anymore." This should be explained in details. From the plot I did not see sandwich structure or not directly.
3. I would like to see how the authors define "sandwich" and "herringbone" structure quantitatively. Now in the phase-V the molecules are still in pair.

manuscript COMMSCHEM-24-0196A

"Polymorphism of pyrene on compression to 35 GPa in a diamond anvil cell"

Reply to Reviewers' comments

Reviewer #1 (Remarks to the Author):

The authors have adequately addressed the referees comments, making the necessary changes throughout the work and appropriately justifying where changes/additions were not made. The purpose and value of the work is now much clearer, and will hopefully be appreciated by the broader crystallographic community.

We thank Reviewer #1 for such positive evaluation of our paper.

Reviewer #2 (Remarks to the Author):

is

I am pleased to report that the authors have successfully addressed all the concerns raised in my initial review. The corrections have significantly improved the quality and clarity of the manuscript. Therefore, I am recommending this revised version for publication.

We thank Reviewer #2 for such positive evaluation of our paper.

Reviewer #3 (Remarks to the Author):

The authors made explanations but did not fully answer my questions, and we did not get new knowledge about the principle/rule of the phase transition of aromatics under high pressure, except for the crystal data.

1. The phase transitions and phase boundaries. The authors need to present some extra experimental evidences to support the phase transitions, or make a calculation on the energy barrier of the phase transitions.

According to the Online dictionary of Crystallography of the International Union of Crystallography (IUCr) https://dictionary.iucr.org/Phase_transition, "Phase transitions of crystallographic interest are those involving solid phases: with the change of the external conditions (temperature, pressure, applied field) the crystal structure undergoes a change that, when it is not accompanied by a change in the chemistry, relates polymorphs". Polymorphism is "The phenomenon in which the same chemical compound exhibits different crystal structures" [Polymorphism - Online Dictionary of Crystallography \(iucr.org\)](https://dictionary.iucr.org/Polymorphism). In our work, using single-crystal X-ray diffraction, we have proven that the crystal structure of pyrene undergoes changes with the change of pressure, thus no extra evidences are needed to claim these phase transitions.

2. The authors wrote "Namely the analysis of the Hirschfeld surfaces and fingerprint plots of pyrene-V gives evidence that this polymorph does not possess sandwich structure anymore." This should be explained in details. From the plot I did not see sandwich structure or not directly.

3. I would like to see how the authors define "sandwich" and "herringbone" structure quantitatively. Now in the phase-V the molecules are still in pair.

Questions 2 and 3 are related, so we answer them together.

In the paper we write: "For visualizing and exploring intermolecular interactions in molecular crystals one uses Hirshfeld surfaces and fingerprint plots. This tool is described in detail in a

comprehensive review by McKinnon et al. (2004) [20]... The Hirshfeld surface itself is defined by the molecule and the proximity of its nearest neighbors, and hence encodes information about intermolecular interactions.” We provide the analysis of Hirshfeld surfaces and fingerprint plots for each pyrene polymorph in the section “Exploring intermolecular interactions using Hirshfeld surfaces and fingerprint plots” following the methodology described previously (see McKinnon et al., 2004, Fabbiani et al, 2006, Spackman & Jayatilaka, 2009, and references therein).

The sandwich-herringbone motif is ubiquitous for PAHs. For the sandwich-herringbone motif it is characteristic that two parallel molecules are arranged in a sandwich motif via π — π stacking interactions, and each motif is arranged in a herringbone packing favouring C—H- π interactions. It means that in a sandwich motif the π — π stacking interactions, manifesting as triangular features on the Hirshfeld surface mapped with shape index, dominate only on one side of the molecules forming pairs (sandwiches), and the front and back views of the Hirshfeld surface look differently (this is the case for pyrene-I, for example; see Fig. 13). If one deals with a simple herringbone packing (https://en.wikipedia.org/wiki/Herringbone_pattern) of molecules, the $\pi \cdots \pi$ stacking interactions appear on both back and front sides, what is the case for pyrene-V. The front and back views of the Hirshfeld surface look quite similar. We agree with Reviewer #3 that crystallographically equivalent molecules in pyrene-V are “still in pair”, but only in the sense that they are related by an inversion centre. They do not form sandwiches. The herringbone pattern has a symmetry of 2D space group *pgg* that is also the case for pyrene-V in projection along the *a* axis.

On request of Reviewer #3, for better clarity we have slightly modified the following statement in the revised manuscript: “In pyrene-V, the molecules are stacked in a simple herringbone motif that immediately reflects in the fingerprint plot and the Hirshfeld surfaces (Figs. 15, S2e,f). The $\pi \cdots \pi$ stacking interactions, manifesting as the alternating red and blue triangles in the Hirshfeld surface mapped with shape index [20], appear now on both back and front sides. Although crystallographically equivalent molecules in pyrene-V are related by an inversion centre, they do not form sandwiches...”

REVIEWERS' COMMENTS:

Reviewer #2 (Remarks to the Author):

[Editorial Note: This reviewer has not provided any further comments to the authors.]